# Patterned human microvascular grafts enable rapid vascularization and increase perfusion in infarcted rat hearts

Meredith A. Redd[1,2,3], Nicole Zeinstra[1,2,3], Wan Qin[1], Wei Wei[1], Amy Martinson[2,3,4], Yuliang Wang[3,5], Ruikang K. Wang[1], Charles E. Murry [1,2,3,4,6] & Ying Zheng[1,2,3]

Vascularization and efficient perfusion are long-standing challenges in cardiac tissue engineering. Here we report engineered perfusable microvascular constructs, wherein human embryonic stem cell-derived endothelial cells (hESC-ECs) are seeded both into patterned microchannels and the surrounding collagen matrix. In vitro, the hESC-ECs lining the luminal walls readily sprout and anastomose with de novo-formed endothelial tubes in the matrix under flow. When implanted on infarcted rat hearts, the perfusable microvessel grafts integrate with coronary vasculature to a greater degree than non-perfusable self-assembled constructs at 5 days post-implantation. Optical microangiography imaging reveal that perfusable grafts have 6-fold greater vascular density, 2.5-fold higher vascular velocities and >20-fold higher volumetric perfusion rates. Implantation of perfusable grafts containing additional hESC-derived cardiomyocytes show higher cardiomyocyte and vascular density. Thus, pre-patterned vascular networks enhance vascular remodeling and accelerate coronary perfusion, potentially supporting cardiac tissues after implantation. These findings should facilitate the next generation of cardiac tissue engineering design.

[1] Department of Bioengineering, University of Washington, Seattle, WA 98109, USA. [2] Center for Cardiovascular Biology, University of Washington, Seattle, WA 98109, USA. [3] Institute for Stem Cell and Regenerative Medicine, University of Washington, Seattle, WA 98109, USA. [4] Department of Pathology, University of Washington, Seattle, WA 98109, USA. [5] Paul G. Allen School of Computer Science & Engineering, University of Washington, Seattle, WA 98109, USA. [6] Department of Medicine/Cardiology, University of Washington, Seattle, WA 98109, USA. These authors contributed equally: Meredith A. Redd, Nicole Zeinstra. Correspondence and requests for materials should be addressed to C.E.M. (email: murry@uw.edu) or to Y.Z. (email: yingzy@uw.edu)

Engineered tissues have emerged as promising approaches to repair damaged organs as well as useful platforms for drug testing and disease modeling[1,2]. However, insufficient vascularization is a major challenge in engineering complex tissues such as the heart[3,4]. Heart failure is the leading cause of death worldwide, and no available treatment options outside of whole heart transplantation address the problem of cellular deficiency[5,6]. Despite this burgeoning clinical need, the therapeutic application of engineered cardiac tissues has not been achieved, partially due to the lack of comprehensive tissue perfusion in vitro and effective integration with host vessels in vivo[4].

Prior efforts to vascularize tissue grafts have mostly relied on self-assembly of endothelial cells (ECs) to form connected tubes within cardiac constructs[7–9]. Although the presence of these vessels improves cardiomyocyte maturation and tissue function, the formed network architecture does not provide efficient perfusion, preventing large-scale construct fabrication and culture. When implanted, these grafts partially integrate with host vasculature but do not establish effective perfusion in a timely fashion[10]. To combat this problem, efforts have been made toward fabricating perfusable vasculature within cardiac tissue constructs in our laboratory and in others[11–13]. Little is known, however, about how these vascular networks will connect with host vessels once implanted and whether physiological systemic perfusion in the grafts can be established.

An engineered tissue also requires appropriate cell sources, which are not only important to promote tissue function but also critical for clinical translation. In particular, the field of vascularization has mostly relied on human umbilical vein endothelial cells (HUVECs), a commonly used endothelial source with known function and availability but poor survival and immunogenic issues in vivo[14,15]. Our laboratory has demonstrated that we can use human pluripotent stem cells to derive ECs (human embryonic stem cell-derived endothelial cells (hESC-ECs))[16,17] and cardiomyocytes[8,18,19] from mesodermal precursors. Importantly, these hESC-ECs exhibit increased angiogenic behavior in flow-derived microphysiological constructs and are vasculogenic when embedded in bulk hydrogel matrix. These properties indicate that hESC-ECs could be an ideal cell source for engineering constructs with high vascular density.

As vascular engineering strategies continue to advance, it is critical to develop better systems to measure perfusion dynamics and achieve more efficient graft–host integration. Standard approaches to assess the graft integration rely on the presence or absence of red blood cells or perfused lectins in histological sections[10]. It has not been possible to directly measure flow and perfusion in the graft and new coronary vasculature. We recently demonstrated an application of optical coherence tomography (OCT)-based optical microangiography (OMAG)[20–24] to obtain high-resolution coronary angiograms on ex vivo Langendorff-perfused and fixed rat hearts[25]. This imaging technique allows for simultaneous image acquisition of high-resolution structural information as well as velocimetry data of the coronary vasculature in both graft and host.

In this study, we combine advanced tissue engineering, stem cell biology, and ex vivo intact heart imaging techniques to study the vascular anastomosis and host integration in the infarcted heart. We demonstrate vascular remodeling and anastomosis in vitro between pre-patterned, perfusable vascular networks and self-assembled (SA) vessels in the bulk matrix, both with hESC-EC cell sources. We show that remodeled constructs with vascular anastomosis have upregulated genes associated with vascular and tissue development. Importantly, these pre-patterned, perfusable constructs improved vascular host integration, which likely supported graft cardiomyocyte remodeling when implanted on an infarcted heart compared to SA controls. Our work demonstrates

that pre-perfused, patterned vessels provide important cues for rapid anastomosis and host integration and sheds light on engineering translational cardiac patches for heart regeneration.

## Results

**Engineering human stem cell-derived microvasculature.** To engineer human stem cell-derived microvessels (μVs) in vitro, we first generated ECs, previously called endocardial-like ECs, from a dual reporter RUES2 human stem cell line (mTmG-2a-puro RUES2[26]) using our recently established differentiation protocol[16,17]. These stem cells stably express either TdTomato red fluorescent protein or (following Cre-mediated recombination) green fluorescent protein (GFP), allowing us to evaluate vascular remodeling between two separately seeded vascular compartments (Supplementary Figure 1A, B). Both mTm- and GFP-expressing hESCs differentiated into endothelial progenitors by differentiation day 5 with >70% populations expressing CD34, a surface marker expressed by both hematopoietic and endothelial populations[27] (Supplementary Figure 1C, D). After an additional week in endothelial culture conditions, endothelial populations arose with >98% purity by day 14 as shown by the expression of endothelial junction protein, CD31 (Supplementary Figure 1E, F). These cells widely express the endocardial transcription factor, NFATC1, throughout the culture (Supplementary Figure 2). We previously demonstrated that these differentiated ECs undergo tubulogenesis when embedded in soft collagen gels and extensive angiogenesis when lined in a three-dimensional (3D) engineered μV platform[16].

Here we further developed constructs with enhanced vascularity by combining the lithographically defined network and SA de novo small tubes of hESC-ECs[28,29]. Microfluidic channel networks were fabricated in collagen gel matrices with and without bulk-seeded GFP-hESC-ECs. The patterned network was then seeded with mTm-hESC-ECs to form the endothelial-lined lumen (Fig. 1a). After culturing under gravity-driven flow for 4 or 7 days, mTm-hESC-ECs in the lumen formed patent μV that retained the original network geometry (Supplementary Figure 3A) and sprouted extensively into the bulk matrix (Fig. 1b, c). GFP-hESC-ECs in the matrix underwent tubulogenesis to form SA lumens in the surrounding collagen (Supplementary Figure 3B). Throughout culture, the ECs retained robust expression of endothelial markers such as vascular endothelial cadherin (VE-cadherin), CD31, and von Willebrand factor (VWF) (Fig. 1c and Supplementary Figure 3C, D). The newly formed SA tubes integrated with angiogenic sprouts from the μV network and formed numerous anastomotic connections (Supplementary Figure 2E and Supplementary Movie 1). Anastomosis was observed between GFP+ de novo tubes and mTm+ endothelium through direct connection at the patterned μV or to the smaller angiogenic sprouts (Fig. 1d, Supplementary Figure 3E). Likewise, many GFP-hESC-ECs did not form de novo tubes, but rather incorporated directly into the vessel wall of the both the sprouts and the pre-patterned μV (Fig. 1d).

To evaluate the effect of de novo lumen formation and anastomosis on the structure and function of the patterned endothelium, we quantified angiogenesis in μVs generated with (μV+SA) and without (μV only) bulk-seeded GFP-hESC-ECs and cultured for 4 or 7 days. While hESC-ECs remodeled extensively and sprouted into the collagen matrix in both groups, the overall number of mTm+ sprouts per vessel surface area was not significantly different between the two groups or from day 4 to day 7 in culture (Fig. 1e). The sprout length and diameter, however, increased over time between 4 and 7 days of culture in μV only constructs, with an overall increasing trend in both sprout length and diameter in the μV+SA constructs over time

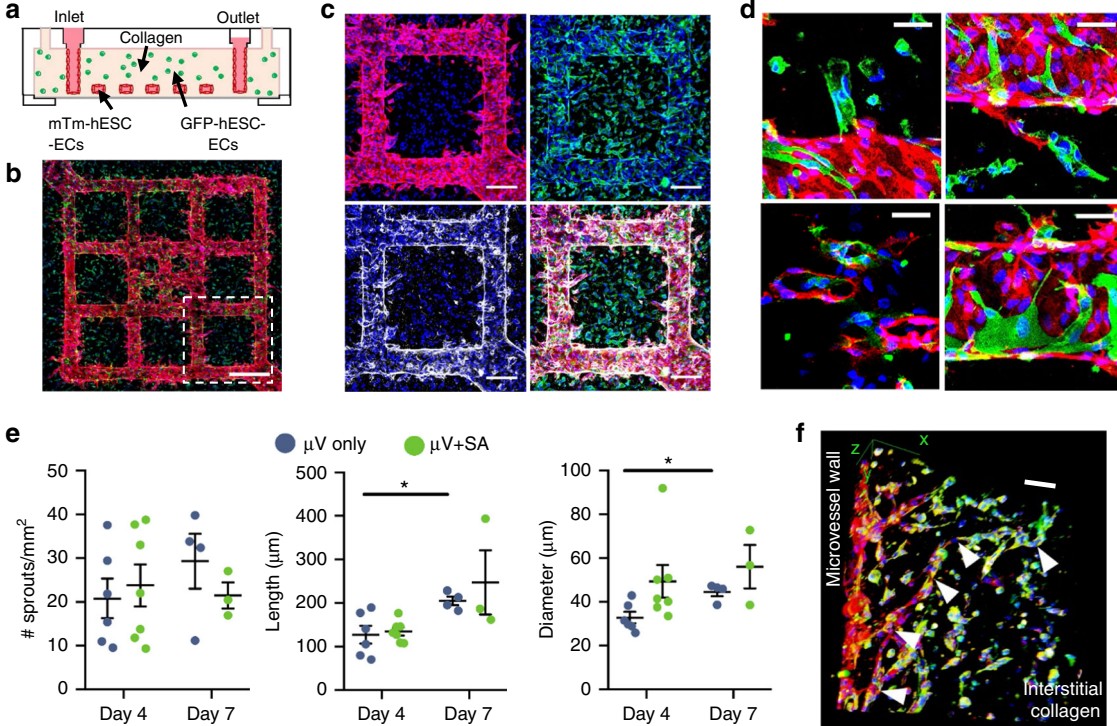

**Fig. 1** In vitro anastomosis of human embryonic stem cell-derived endothelial cells (hESC-ECs) in engineered microvessels (μVs). **a** Schematic of in vitro culture device for μV+SA constructs: mTm-hESC-EC μVs formed via perfusion and attachment with bulk-seeded GFP-hESC-ECs in the surrounding collagen gel. **b** Maximum intensity projection of stitched large image confocal z-stack of μV+SA construct cultured for 4 days and stained for DsRed (red) and GFP (green) to detect mTm- and GFP-expressing hESC-ECs, respectively. Scale bar, 500 μm. **c** Outlined region (white box) in **b** stained for DsRed (red, top left), GFP (green, top right), and VE-cadherin (white, bottom left). Merged image, bottom right. Scale bar, 200 μm. **d** High magnification images of GFP-hESC-ECs (green) integrated with mTm-hESC-EC (red) patterned vessel in μV+SA constructs. Scale bar, 50 μm. **e** Quantitation of sprouts from patterned μVs by sprout density (no. of sprouts per vessel surface area), sprout length, and sprout diameter in μV only (blue circles) and μV+SA (green circles) constructs after 4 days and 7 days of culture. N = 6, 7, 4, and 3 biologically independent samples for D4 μV only, D4 μV+SA, D7 μV only, and D7 μV+SA, respectively. p = 0.011 for length and p = 0.007 for diameter for D4 μV only and D7 μV only, p > 0.05 for all others (two-tailed t test). **f** 3D view of GFP+ de novo lumen integrated with mTm+ microvascular sprout (white arrowheads) stained for CD31 (red) and GFP (green). Scale bar, 100 μm. Representative images for **b–d**, **f** from seven biologically independent samples of D4 μV+SA, with similar results. Hoechst-stained nuclei, blue. Error bars, mean ± SEM. *p < 0.05 determined using two-tailed t test. D4 after 4 days of culture, D7 after 7 days of culture

and compared to μV only (Fig. 1e). This suggests that the interstitial de novo tubes help establish and stabilize endothelial sprouts from the patterned μVs. 3D rendering of confocal z-stacks further confirmed the extensive integration and anastomosis between GFP+ lumens near the patterned, perfused network and the microvascular spouts that penetrated deep into the interstitial collagen (Fig. 1f).

We next analyzed de novo lumen density and average lumen size with respect to the distance from the μV wall in μV+SA constructs. GFP+ lumen density was significantly decreased with increased distance from the vessel wall after 7 days in culture with a similar trend after just 4 days (Supplementary Figure 4A-D). The density of GFP+ cells near the vessel wall was comparable from day 4 to day 7 suggesting that the decline in EC density at the larger distances (>600 μm) was likely due to lumen regression or cell death rather than migration (Supplementary Figure 4D). Ethidium homodimer-1 staining showed trends toward decreasing cell viability with increased distance from the vessel wall and was lower at day 7 compared to day 4 (Supplementary Figure 4E). The average size of GFP+ lumens near the vessel wall (within 300 μm) was significantly larger than lumens located >600 μm distance away after 4 and 7 days of culture (Supplementary Figure 4F). These findings suggest that perfusion promotes better de novo lumen formation, which in turn leads to better remodeling and anastomosis.

**In vitro perfusion of μVs and angiogenic sprouts.** We examined the flow and perfusion characteristics of engineered μVs and endothelial sprouts in μV only and μV+SA constructs. Fluorescent beads, perfused from the vessel inlets, were observed to immediately fill the endothelial-lined, patterned microchannels before moving into the endothelial sprouts (Supplementary Movie 2 and Fig. 2a). By comparing bead perfusion images with immunofluorescent confocal images, we confirmed that sites of anastomosis between the de novo lumens and the patterned μV were patent and perfusable (Fig. 2a, b). The total perfused area of the constructs increased with time in both μV+SA and μV only groups but was not significantly different between the two groups (Supplementary Figure 5A-C). This suggests that substantial vascular remodeling occurred over time, but the addition of perfusable anastomotic connections in μV+SA constructs was not yet sufficient to influence the global perfusion dynamics of the constructs. There was no significant difference in flow velocity in the pre-patterned channel (diameter >100 μm) in both vessel conditions (Fig. 2c). In sprouts with diameters <50 μm, the average bead velocity was significantly higher in μV+SA (87.9 ± 5.6 μm/s, mean ± standard error) compared to μV only constructs (42.6 ± 8.4 μm/s) after 7 days of culture (Fig. 2d), whereas differences in day 4 constructs were not significant. An upward shift in sprout velocity was observed across all diameters between 10 and 50 μm in day 7 μV+SA constructs (Fig. 2e). Flow simulation

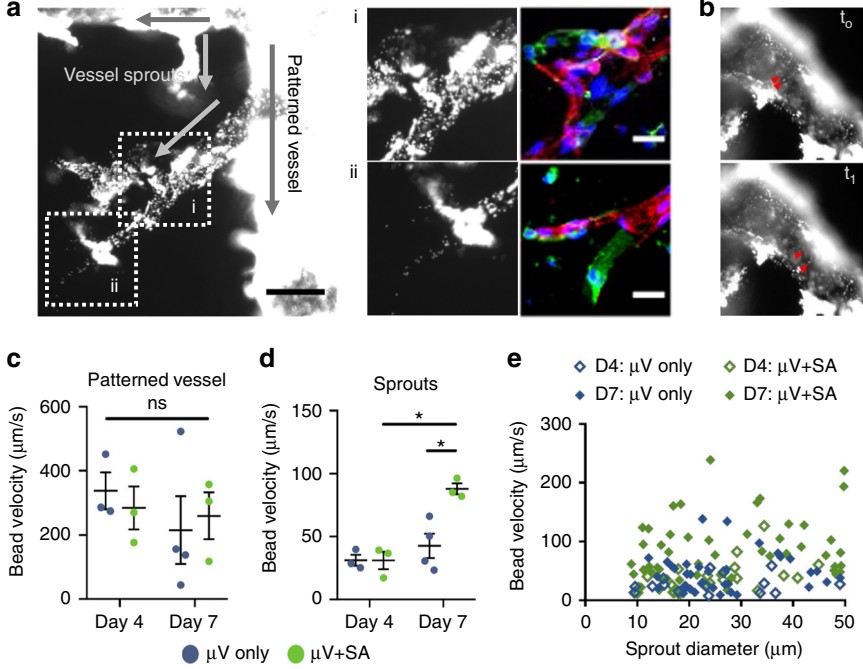

**Fig. 2** Fluorescent bead perfusion of engineered microvessels (μVs). **a** μV+SA construct with perfused anastomotic connections. High magnification views of outlined regions i and ii with corresponding in situ staining for mTm-hESC-ECs (DsRed+, red) and GFP-hESC-ECs (GFP+, green). Scale bars, 100 μm, 40 μm. **b** High magnification view of fluorescent beads at two time points, $t_0$ and $t_1$, 0.27 s apart. Red arrows track the movement of fluorescent beads to calculate velocities. **c, d** Average bead velocity in the patterned vessel (**c**) and in sprouts with diameter <50 μm (**d**) in μV only (blue circles) and μV+SA (green circles) constructs after 4 days and 7 days of culture $N$ = 3, 3, 4, and 3 biological independent samples for D4 μV only, D4 μV+SA, D7 μV only, and D7 μV+SA, respectively. $p$ = 0.004 for D4 μV+SA and D7 μV+SA in sprouts, $p$ = 0.012 for D7 μV and D7 μV+SA in sprouts, $p$ > 0.05 for all others (two-tailed $t$ test). **e** Scatter plot of bead velocity in relation to sprout diameter $N$ = 15, 20, 37, 53 (number of sprouts analyzed) for D4 μV only, D4 μV+SA, D7 μV only, and D7 μV+SA in 2, 1, 5, and 3 biologically independent samples. Representative images for **a**, **b** from three biologically independent samples of D4 μV+SA, with similar results. Hoechst-stained nuclei, blue. Error bars, mean ± SEM. *$p$ < 0.05 determined using two-tailed $t$ test. ns non-significant ($p$ > 0.05). D4 after 4 days of culture, D7 after 7 days of culture

of idealized sprouted vessel networks showed that the addition of sprouts led to higher flow in the vessel networks when the same pressure drop is applied between an inlet and an outlet, indicating a lower flow resistance (Supplementary Figure 5D). Together, these data suggest that localized vascular remodeling and anastomotic events decrease the vascular resistance, allowing more efficient perfusion through endothelial sprouts.

**Stem cell-derived μVs are non-thrombogenic.** Next, we investigated the interaction between blood and hESC-EC-derived engineered μVs. Citrate-stabilized, ABO-matched whole blood with labeled platelets was perfused from the inlet and into the μV lumens. Most blood flowed freely and continuously through the patterned μV and exited from the outlet throughout 20–30 min of perfusion (Supplementary Movie 3 and Fig. 3a). Some blood cells entered the endothelial sprouts (Supplementary Movie 3) and eventually stopped, presumably due to their dead-end architecture. In regions where sprouts connected two branches of the patterned vessel, individual red blood cells could be seen passing through the newly formed anastomosis bridges (Supplementary Movie 4). Few red blood cells clumped or adhered to the vessel throughout the blood perfusion, and these could be completely washed out of the main vessel without obstructing flow (Supplementary Figure 6A). Small amounts of platelets, labeled with antibodies to CD41a, aka platelet-specific glycoprotein IIb, adhered to the vessel wall of engineered μVs but without formation of large aggregates (Supplementary Movies 5 and 6). Platelet adhesion remained at a low level throughout the blood perfusion over 20 min (Supplementary Figure 6B), although there was somewhat greater adhesion in the hESC-EC constructs than

in quiescent HUVEC-seeded μVs[28]. Subsequent immunofluorescent analysis further confirmed low platelet adherence on the walls of hESC-EC endothelium, and platelets (CD41a) that were adhered were primarily localized to endothelial junctions (CD31) (Fig. 3b, c, Supplementary Figure 6C, D). When activated with phorbol myristate acetate (PMA) or interleukin (IL)-1β, the hESC-EC μVs showed increased platelet adhesion after 20 min of blood perfusion (Fig. 3b, c, Supplementary Figure 6E, F). These findings demonstrated that hESC-EC-formed μVs are non-thrombogenic and can convert to a thrombogenic state as a physiological response to stimuli.

**Unique μV gene expression of vascular development.** To better understand the difference among SA only, μV only, and μV+SA constructs, we collected RNA for transcript profiling using RNAseq analysis for the three groups after 3 days of culture in vitro. The μVs in the μV only and μV+SA constructs were made from a large grid pattern (13 × 13) with lumen diameter of 125 μm to maximize the vascular surface and RNA yield. Although we predicted that the μV+SA constructs would have an intermediate profile between the μV only and the SA only constructs, this was not the case. Strikingly, >5000 genes were significantly different (fold change >1.5 and false discovery rate (FDR) < 0.05) in μV+SA constructs compared to SA constructs, whereas approximately 500 genes were significant when comparing μV vs. SA constructs. We performed principal component analysis (PCA), which showed that PC1 separated the μV+SA constructs from the other two, whereas in PC2 the μV+SA constructs were intermediate between the μV only or the SA only constructs (Fig. 4a). The top genes that differentiated the μV+SA

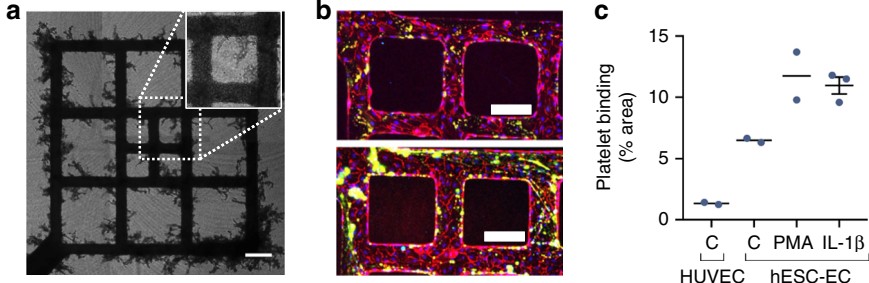

**Fig. 3** Citrated whole-blood perfusion in engineered microvessels (μVs). **a** Brightfield stitched large image of red blood cell-filled pattern and sprouts with magnified view (inset, white dotted boundary) for human embryonic stem cell-derived endothelial cell (hESC-EC)-seeded μV only constructs after 4 days of culture. Scale bar, 200 μm. **b** Maximum intensity projection of confocal z-stack of constructs with adhered platelets after 30-min perfusion and subsequent phosphate-buffered saline (PBS) washes for untreated hESC-EC-seeded constructs (top) and phorbol myristate acetate (PMA)-treated hESC-EC-seeded constructs (bottom) stained for CD31 (red) and CD41a (green). Scale bar, 200 μm. **c** Quantification of platelet adhesion on the vessel wall for constructs seeded with human umbilical vein endothelial cells (HUVECs) in control conditions (C–HUVEC), hESC-ECs in control conditions (C–hESC-EC), and hESC-ECs treated with PMA (PMA–hESC-EC) or interleukin (IL)-1β (IL-1β–hESC-EC). Data are expressed as a percentage of the vessel wall surface area. $N = 2, 2, 2$, and 3 biologically independent samples for C–HUVEC, C–hESC-EC, PMA, and IL-1β, respectively. Representative images for **a**, **b** from two biologically independent samples of C–hESC-ECs and two biologically independent samples of PMA–hESC-EC, with similar results. Error bars, mean ± SEM

constructs from the other two in PC1 (Fig. 4b) include enzymes such as *ENO2*, *ALDOA*, and *ARG2*; peptidases such as *MME* and *MMP9*; glucose transporters such as *SLC2A1* (aka *GLUT1*) and *SLC2A3* (aka *GLUT3*); growth factors such as *VEGFA* and *VEGFB*; and other angiocrine factors such as *ANGPTL4* and *IGFBP5* (Fig. 4c).

Next, we performed Differential Expression Analysis (Supplementary Figure 7A) and Gene Ontology terminology analysis (Fig. 3d) which showed that μV+SA, compared to SA condition, had significant upregulation in hypoxia, glycolysis, tumor necrosis factor-α signaling via nuclear factor-κB, mammalian target of rapamycin C1 (mTORC1) signaling, and epithelial–mesenchymal transition (EMT). The comparison of μV vs. SA showed similar patterns, though to a lower extent and with fewer significant genes (Fig. 4d and Supplementary Figure 7B). Canonical pathway analysis showed that the μV+SA constructs had significant upregulation of EIF2 signaling, mTOR signaling, IL-8 signaling, EMT regulation, glycolysis, hypoxia-inducible factor-1α signaling, as well as signaling of vascular endothelial growth factor (VEGF), endothelin-1, Wnt/β-catenin, Neuregulin, and C-X-C chemokine receptor 4, and downregulation of phosphatase and tensin homolog, Notch, and apoptosis signaling (Supplementary Figure 7C). The functional annotation associated in these analyses showed increased function in development of endothelial tissue, EC proliferation, cell assembly, and DNA binding and interactions. These expression analyses suggest that μV+SA constructs are significantly different compared to SA only or μV only ones in terms of endothelial phenotypes and vascular development, which may change their host integration in vivo as well.

**Patterned vascular graft implantation into infarcted rats**. To identify whether patterning and perfusion in the constructs could accelerate vascular integration with host myocardium, we implanted both perfusable (μV+SA) and non-perfusable (SA) constructs onto the epicardial surfaces of infarcted athymic nude Sprague-Dawley rat hearts 4 days after ischemia/reperfusion (I/R). Both types of constructs were 8 mm in diameter with 1 mm thickness and contain similar amounts of ECs (μV+SA: 150,000 cells; and SA: 140,000 cells) (Supplementary Figure 8A). A large grid pattern (13 × 13), same as that used in RNAseq studies, was used for μV+SA constructs (Supplementary Figure 8A). Both types of vascular constructs were cultured for 4 days to allow for tubulogenesis in bulk matrix and vascular remodeling, as characterized above, and then sutured onto the epicardium of the left

ventricle (Supplementary Figure 8B). To detect early vascular integration, the rats were euthanized 5 days after the construct implantation, and their hearts were arrested in diastole and excised followed by perfusion fixation at physiological pressures (~100 mm Hg). Since most coronary blood flow occurs during diastole[30], arresting the heart in this state prior to its excision allowed for the study of coronary vascular perfusion at its maximal capacity. Upon excision, the grafts were easily identifiable on the epicardial surface of intact hearts as well as in post-processed histological sections (Supplementary Figure 8B, C). Importantly, the infarct size was comparable between groups and in the range of previous rat I/R studies in our laboratory[31,32] (Supplementary Figure 8D), and similar inflammatory responses were observed as evaluated by macrophage (CD68+) infiltration to injured and graft regions (Supplementary Figure 8E-F).

**Graft vascular perfusion by OMAG**. To determine the extent of vascular integration between the rat myocardium and the implanted vascular grafts, we performed ex vivo real-time imaging of graft perfusion prior to histological processing of the hearts. OMAG scans of the coronary vasculature were collected during Langendorff perfusion. The coronary flow was driven by an aortic perfusion pressure of 90 mm Hg, which is within the range of normal diastolic aortic pressures for both healthy rat and human hearts[33,34]. Two OMAG imaging protocols, OMAG and OMAG- based capillary velocimetry (OMAG-V) (see Methods), were used in this study in order to acquire high-resolution images of vascular structure and quantitative blood velocities within the grafts. In normal healthy regions of the heart, OMAG and OMAG-V images revealed dense vasculature with hierarchical branching structure and high flow rates in large arteries when compared to smaller vessels (Fig. 5a). μV only constructs showed minimal vascular flow at 5 days after implantation (Supplementary Figure 9A), and therefore, these were not included in the velocity measurements. Considerable vascular flow was observed in the pre-patterned, perfusable vascular grafts (μV+SA) with higher velocities in larger diameter vessels, whereas the SA grafts had few visible perfused vessels (Fig. 5b, c). Since OMAG data acquisition and processing uses sequential frames to distinguish moving from stationary particles, a single 3D scan gives rise to both structural information and vascular flow. To distinguish between host and graft, OMAG flow data were overlaid with structural information extracted from the same image scan (Supplementary Movies 7 and 8), and 3D images were generated

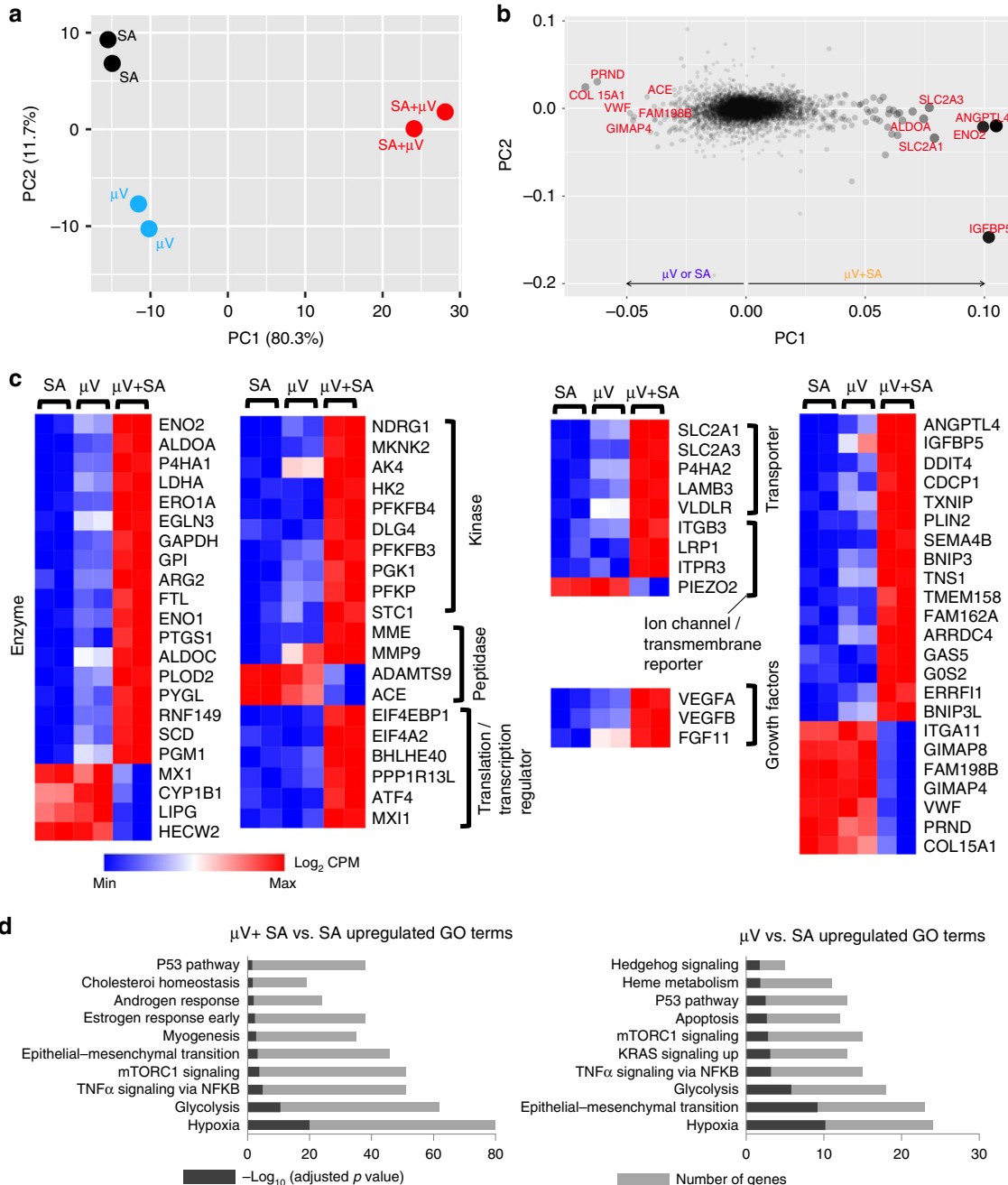

**Fig. 4** Global RNA sequencing reveals differential gene expression profiles among self-assembled (SA), μV only, and μV+SA constructs after 3 days of culture in vitro. **a** 2D principal component analysis (PCA) of RNA sequencing data for cultured constructs showing clustered groups for each condition. **b**, **c** Top genes contributing to PC1: **b** Size of the circle is proportional to the contribution. **c** Heat map of the top 75 genes (log₂ CPM) contributing to PC1 in categorized functions. Colormap normalized to minimum and maximum expression. Red, high expression. Blue, low expression. **d** Gene ontology terminology analysis (GO) showing different gene clusters for μV+SA vs. SA (left) and μV vs. SA (right). Each sample is two constructs pooled. CPM counts per million

(Fig. 5d). This overlay reveals drastic improvement in vascular perfusion of μV+SA grafts compared to SA grafts. Quantification of the overall vessel area density in OMAG images confirmed that μV+SA grafts had significantly more perfused vessels (33.8 ± 5.6% of tissue area) compared to SA grafts (5.3 ± 1.7% of tissue area). Vessel area density in neighboring non-infarcted regions of both groups was comparable at 63.5 ± 9.2% and 64.5 ± 5.1% for SA and μV+SA grafts, respectively (Fig. 5e). Perfusion dynamics of individual vessels in the grafts were assessed by using the linear correlation between velocity and signal intensity in OMAG-V scans (Supplementary Figure 9B)[35]. The average measured

velocity (vessel diameter between 20 μm and 40 μm) in μV+SA grafts (0.72 ± 0.09 mm/s) was similar to that of the non-infarcted myocardium and significantly higher than that of SA grafts (0.29 ± 0.10 mm/s) (Fig. 5f). The perfusion rate for the field of view was 24.5 ± 8.8 μL/min in μV+SA grafts, corresponding to a volumetric perfusion rate of approximately 12.3 ± 4.4 mL/min/mL tissue. This perfusion rate is >20-fold greater than that of the SA grafts which had a perfusion rate of 0.9 ± 0.8 μL/min (0.5 ± 0.4 mL/min/mL tissue) (Fig. 5g). Despite this improvement, perfusion rates in both grafts were significantly less than rates in non-infarcted healthy regions of the same hearts (117.9 ± 32.6 μL/min

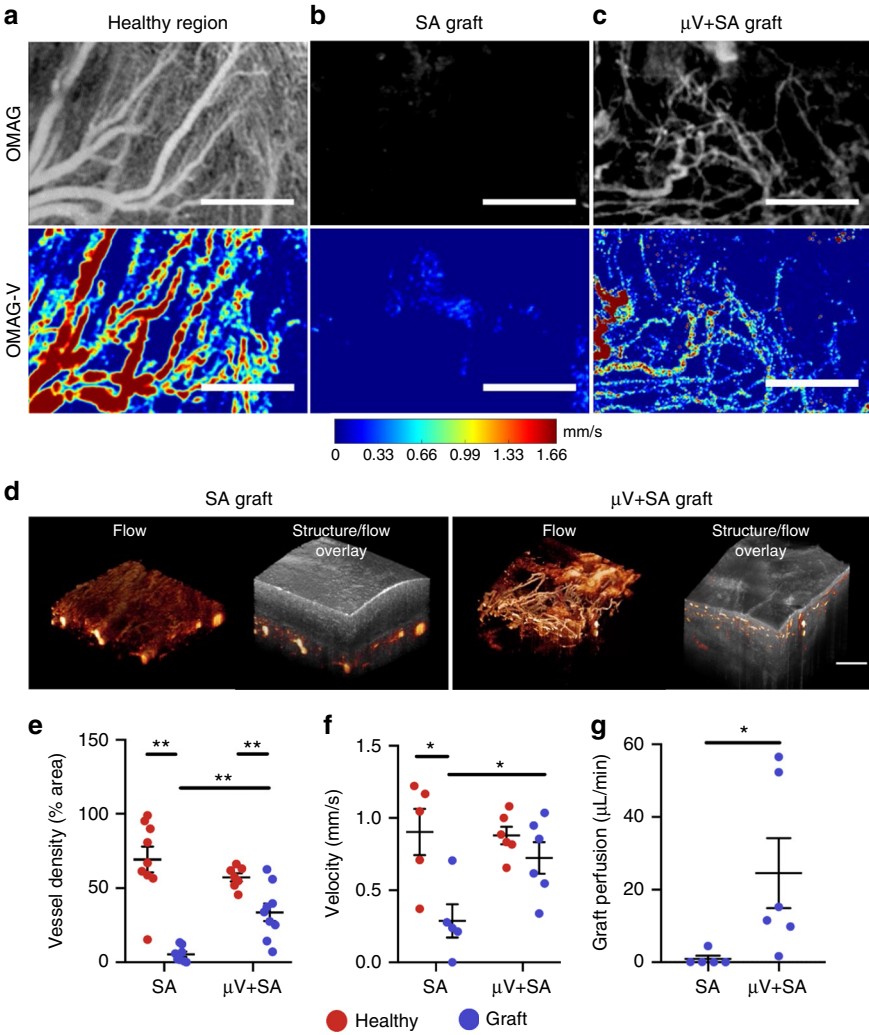

**Fig. 5** Implanted vascular constructs in infarcted rat heart model with optical microangiography (OMAG) assessment of real-time perfusion 5 days post-implantation. **a–c** Doppler-based images of vascular flow in Langendorff perfused engrafted hearts for **a** healthy region, **b** control graft (self-assembled (SA)), and **c** patterned, perfusable graft (μV+SA). Top panels, high-resolution, non-quantitative OMAG images. Bottom panels, OMAG-V images with linear correlation between intensity and velocity. Scale bar, 500 μm. **d** Three-dimensional views of OMAG flow data and overlay of OMAG structure and flow data for SA graft and μV+SA graft. Scale bar, 500 μm. **e** Quantitation of vessel density of perfused vessels in healthy (red circles) and graft (purple circles) regions for SA and μV+SA. $N = 9$ and 8 biologically independent animals for SA and μV+SA, respectively. $p = 0.0009$ for SA healthy and graft, $p = 0.0016$ for μV+SA healthy and graft, $p = 0.0012$ for SA and μV+SA graft, $p > 0.05$ for all others (two-tailed $t$ test). **f** Velocimetry measurements of blood flow velocity in small arterioles (diameter between 20 and 40 μm) of healthy and graft regions. $N = 5$ and 6 biologically independent animals for SA and μV+SA, respectively. $p = 0.016$ for SA healthy and graft, $p = 0.023$ for SA and μV+SA graft, $p > 0.05$ for all others (two-tailed $t$ test). **g** Volumetric perfusion rate in SA and μV+SA grafts from velocimetry data. $N = 5$ and 6 biologically independent animals for SA and μV+SA, respectively. $p = 0.038$ for SA and μV+SA graft (two-tailed $t$ test). Representative images for **a–d** from 5 biologically independent animals containing SA grafts, 6 biologically independent animals containing μV+SA grafts, and 11 corresponding healthy regions from all animals containing grafts, with similar results. Error bars, mean ± SEM. *$p < 0.05$, **$p < 0.01$ determined using two-tailed $t$ test

(healthy region of SA), 113.9 ± 16.9 μL/min (healthy region of μV+SA)). These results demonstrate that engineered vascular constructs integrate functionally with the host vasculature and that pre-patterned, perfusable vasculature improves graft perfusion dynamics.

**Human μVs survived implantation and were perfused**. To identify and distinguish between perfused human vessels vs. infiltrating host vessels in the grafts, we perfused lectins through the ex vivo rat hearts prior to histological processing. Two types of lectins were retrograde perfused: fluorescein *Griffonia simplicifolia* Lectin I (GSL I) and rhodamine *Ulex europaeus* Agglutinin I (UEA I) to label rat and human endothelium, respectively. A non-specific lectin antibody was then used to recognize all

perfused vessels (of both rat and human origin), while an anti-rhodamine antibody was used to specifically detect UEA I+ perfused human vessels within the graft. Antibodies against TdTomato (anti-DsRed) identified all human ECs (Fig. 6a, b, Supplementary Figure 10A). The presence of human vessels was confirmed with additional staining for human endothelial junction protein, hCD31 (Supplementary Figure 10B-D). Quantification of the density and size of hESC-EC vessels in the grafts showed no difference between μV+SA and SA groups (Fig. 6c). The density of total perfused vessels (lectin+), however, was greater in μV+SA constructs (373 ± 94 vessels/mm²) compared to SA grafts (149 ± 55 vessels/mm²), along with an increasing trend in the average vessel size (Fig. 6c). The number of perfused vessels in grafts is still less than the host infarct region

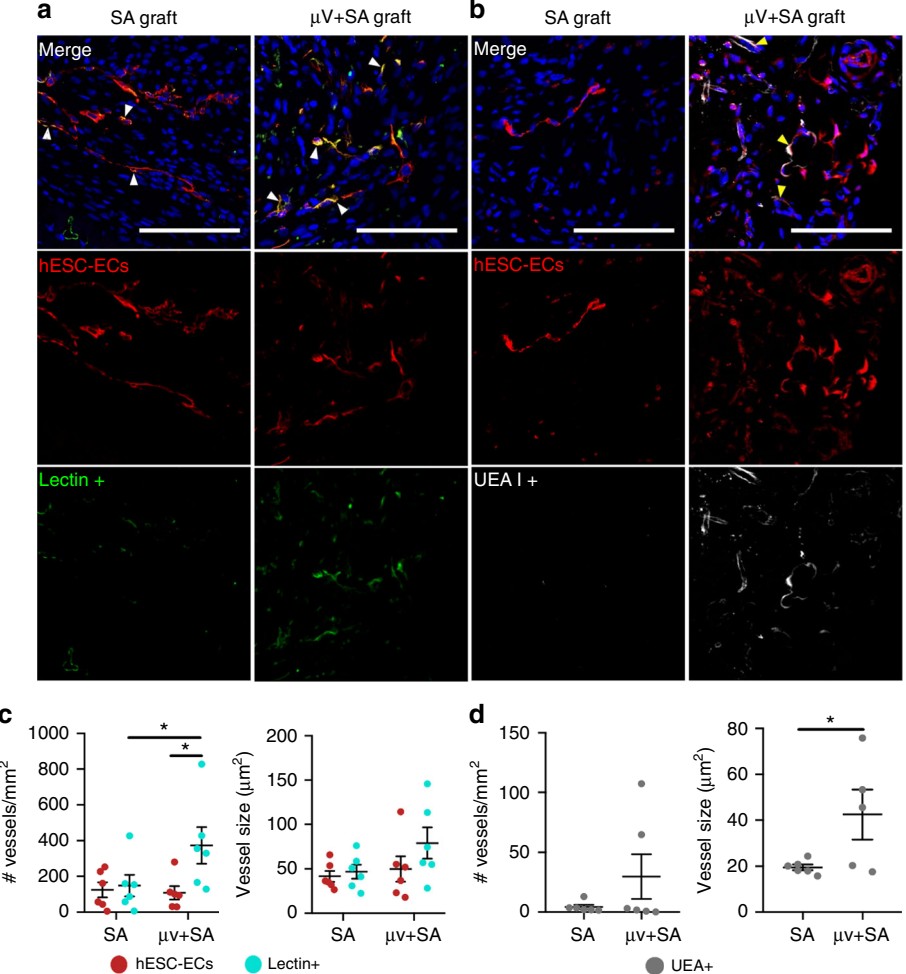

**Fig. 6** Detection of human endothelial cells and perfused vessels in grafts 5 days post-implantation. **a, b** Immunofluorescent-stained paraffin sections of self-assembled (SA) and μV+SA grafts: **a** human embryonic stem cell-derived endothelial cells (hESC-ECs) (DsRed+, red) and perfused rat and human endothelium (Lectin+, green). White arrows in merged images denote double positive cells. Scale bar, 100 μm. **b** hESC-ECs (DsRed+, red) and perfused human endothelium (UEA I+, white). Yellow arrows in merged images denote double positive cells. Scale bar, 100 μm. **c** Quantification of vessel density (number of vessels per mm$^2$) and average vessel size for hESC-EC vessels (DsRed+, red circles) and all perfused vessels (Lectin+, blue circles) in the grafts. $N = 6$ biologically independent animals for both SA and μV+SA grafts. $p = 0.045$ for lectin+ vessels for SA and μV+SA, $p = 0.018$ for DsRed+ and lectin+ for μV+SA, $p > 0.05$ for all others (two-tailed $t$ test). **d** Quantification of vessel density and average vessel size of perfused human vessels (UEA I+, grey circles) in the grafts. $N = 6$ biologically independent animals for both SA and μV+SA grafts, with one animal with μV+SA graft excluded from vessel size calculation due to lack of UEA I+ vessels. $p = 0.023$ for vessel size for SA and μV+SA, $p > 0.05$ for vessel density (two-tailed $t$ test). Representative images for **a**, **b** from six biologically independent animals containing SA grafts and six biologically independent animals containing μV+SA grafts, with similar results. Hoechst-stained nuclei, blue. Data were collected from at least three confocal images of randomly selected regions per sample and analyzed by a custom lumen identifying code for vessel density and size. Error bars, mean ± SEM. *$p < 0.05$ determined using two-tailed $t$ test. UEA *Ulex europaeus* Agglutinin I

(Supplementary Figure 10E). To specifically identify human vessels that integrated with host circulation, we quantified UEA1 + vessels, which was higher in μV+SA grafts (29.6 ± 17.0 vessels/ mm$^2$) compared to SA grafts (4.2 ± 1.6 vessels/mm$^2$). The fact that the perfused human vessels were 10% of the total perfused vessels indicates that most of graft perfusion was due to host vascular ingrowth (Fig. 6d). Smooth muscle encoating within graft human vessels was observed, but in rare events in vascular grafts, and did not appear to be different between groups (Supplementary Figure 10B–D). These findings confirm histologically that our composite perfusable vascular grafts are better perfused and integrated with host coronary circulation than SA non-perfusable grafts and that the implanted human ECs contributed to this effect.

**Pre-patterned cardiac grafts survived implantation and were perfused.** To determine the benefit of increased early perfusion of engineered μV grafts, we implanted both perfusable (μV+SA) and non-perfusable (SA) cardiac constructs containing additional hESC-derived cardiomyocytes (hESC-CMs) onto infarcted athymic Sprague-Dawley rat hearts using the previously described procedure for vascular graft implantations. The cardiac constructs contained hESC-CMs and human bone marrow-derived stromal cells HS-27a in addition to hESC-ECs at a ratio of 4:1:1, respectively, in bulk matrix. hESC-ECs were also seeded and cultured under perfusion within the pre-patterned lumens as described earlier. All hESC-CM populations used for cardiac constructs had cardiomyocyte purity >70% as indicated by the expression of cardiac marker cTnT (Supplementary Figure 11A,

B). After 4 days of culture, spontaneous beating was detected in both µV+SA and SA cardiac constructs (Supplementary Movies 9 and 10). At 5 days post-implantation, hearts were excised, and histological analysis was performed. Both groups had similar infarct sizes (Supplementary Figure 11C-D). To assess hESC-CM graft retention, we stained for human-specific β-myosin heavy chain (β-MHC) (Fig. 7a, b). The density of β-MHC+ cells was significantly greater in the µV+SA cardiac grafts (194 ± 31 cells/mm$^2$) than the SA grafts (85 ± 12 cells/mm$^2$). The β-MHC+ graft size, normalized by the left ventricle size, was measured to be greater, however, not significantly, in µV+SA grafts than SA grafts (0.16 ± 0.05% and 0.11 ± 0.02% respectively; $p = $ NS, two-tailed $t$ test) (Fig. 7c). This suggests that perfusable µVs enhanced hESC-CM remodeling, which could potentially improve cardiac engraftment and function long term. The total perfused vessel density was also significantly greater in µV+SA cardiac grafts (320 ± 53 vessels/mm$^2$) than in SA cardiac grafts (179 ± 24 vessels/mm$^2$) (Fig. 7d). Additionally, the average lumen size was greater, though not significantly, in µV+SA grafts (75 ± 32 µm) than in SA grafts (40 ± 8 µm) (Fig. 7d). We next sought to determine whether cells within the grafts were viable and performed a terminal deoxinucleotidyl transferase-mediated dUTP-fluorescein nick end labeling (TUNEL) assay to mark apoptotic cells (Fig. 7e). Both µV+SA and SA grafts had low levels of apoptotic cells (1.1 ± 0.5% and 1.9 ± 0.5%, respectively; $p = $ NS, two-tailed $t$ test; Fig. 7f). To specifically determine hESC-CM viability, we performed a TUNEL assay with an α-actinin co-stain to label cardiomyocytes. We were unable to identify any TUNEL +/α-actinin+ cells, indicating that remaining cardiomyocytes had high viability and possibly passed the peak cell death at 5 days post-implantation[36] (Supplementary Figure 11E). These findings demonstrate the beneficial role of a perfusable, patterned vascular network in cardiac graft remodeling, which could lead to enhanced engraftment and function long term.

## Discussion

Vascularization and rapid host integration remain as critical challenges for engineered cardiac tissues and for their successful translation to the clinic for heart regeneration. While considerable progress has been made to generate pre-vascularized tissues, efficient perfusion has not been achieved in vitro or in vivo when implanted on injured myocardium. In this study, we successfully engineered highly vascularized constructs from stem cell-derived ECs by incorporating SA capillary tubes around a patterned and perfusable microvascular network. We demonstrated that pre-patterned, perfusable µVs underwent extensive remodeling and anastomosis with de novo capillary tubes in vitro, and the resulting perfusable vascular construct systemically integrated with host vasculature after implantation better than non-patterned, non-perfusable constructs.

Our study demonstrated the successful use of hESC-ECs and potentially patient-derived induced pluripotent stem cells as the vascular cell source in engineered tissue constructs. The use of a patient-specific autologous cell source would bypass the immunogenic risks associated with clinical transplantation[14]. In addition, the use of ECs at the same developmental stage as the cardiomyocytes may provide important signaling to promote cardiomyocyte maturation and tissue function. Our work took a step beyond EC differentiation from hESCs and further built 3D µVs that allow for robust culture and phenotype maintenance under flow. We demonstrated and utilized their angiogenic, tubulogenic, and nonthrombogenic properties toward engineering highly vascularized constructs for tissue engineering applications.

Our studies uncover that perfusion facilitates anastomosis and vascular remodeling in vitro. We showed that anastomosis occurs between de novo capillary tubes and pre-patterned vascular conduits and that bulk-seeded ECs directly incorporate into the microvascular endothelium, leading to increased flow velocity and decreased vascular resistance. This phenomenon appears similar to the incorporation of endothelial progenitor cells into active sites of angiogenesis in animal models of ischemia[37]. Transcriptional analysis further revealed unique gene clusters toward upregulated signaling in hypoxia, glycolysis, and vascular development in anastomosed constructs. These studies demonstrate the formation of a highly remodeled perfusable vascular network with numerous anastomotic connections, and more importantly, these vessels are capable of carrying blood without blockage.

Our implantation studies quantitatively demonstrate that perfusable vascularized grafts integrated significantly better than non-perfusable SA vascular grafts. We exploited OCT-based imaging technology in the intact heart to assess the perfusion dynamics in grafts under physiological pressure, which had not been achieved previously. The live heart poses a major challenge for assessment of coronary flow due to large motion artifacts. Our use of diastolically arrested, fixed hearts allowed for precise control of the applied pressure at physiological level for flow measurement while eliminating the motion artifacts associated with the beating heart. While this is admittedly different from a living, beating heart, it does present a snapshot of the heart's microcirculation at the peak phase of coronary perfusion. This, in combination with intralipid perfusion to mimic blood flow and generate angiograms, made it possible to obtain the 3D vascular structure, flow velocity, and perfusion rate of the vasculature down to the capillary scale in tissue grafts implanted on infarcted hearts. More importantly, our approach allowed for unambiguous distinction between the host and graft with respect to their tissue structure and perfusion dynamics. Our patterned vascular grafts at 5 days post-implantation demonstrated comparable perfusion velocities (0.72 mm/s in 20–30 µm sized vessels) to those in the non-infarcted myocardium as well to previously reported values in similar sized venules and arterioles in living tissues[38–40].

The ability to quantify the perfusion dynamics of implanted vascular grafts ex vivo could provide a more precise readout for comparing different vascularization strategies to achieve better transport efficiency, which would benefit future development of vascular engineering techniques. One limitation to this approach is the inability to distinguish human vessels from rat vessels during the real-time OMAG data acquisition. In this study, we relied on subsequent histological detection to confirm the role of implanted human cells, which limits the information that can be gathered on the extent the cells within the patterned network contributed to the increased perfusion dynamics. A system that has dual imaging capabilities with both OCT and fluorescent-based imaging could potentially be used to overcome this challenge.

Despite a normal range of velocities displayed in vessels within patterned vascular grafts, these newly integrated vessels do not yet structurally resemble proper coronary structure. The graft vessels are more sparsely distributed with less organized structures compared to the dense, aligned vasculature displayed by their healthy counterparts. Surprisingly, the overall structure of perfused graft vessels did not retain the original geometry of the network pattern, but the pre-patterning did enhance vascular density and perfusion. Our histological assessment suggested that, although the implanted human ECs did contribute to the newly connected and perfused vessels in the graft, most perfused vessels originated from the host. This suggests that the pre-patterning

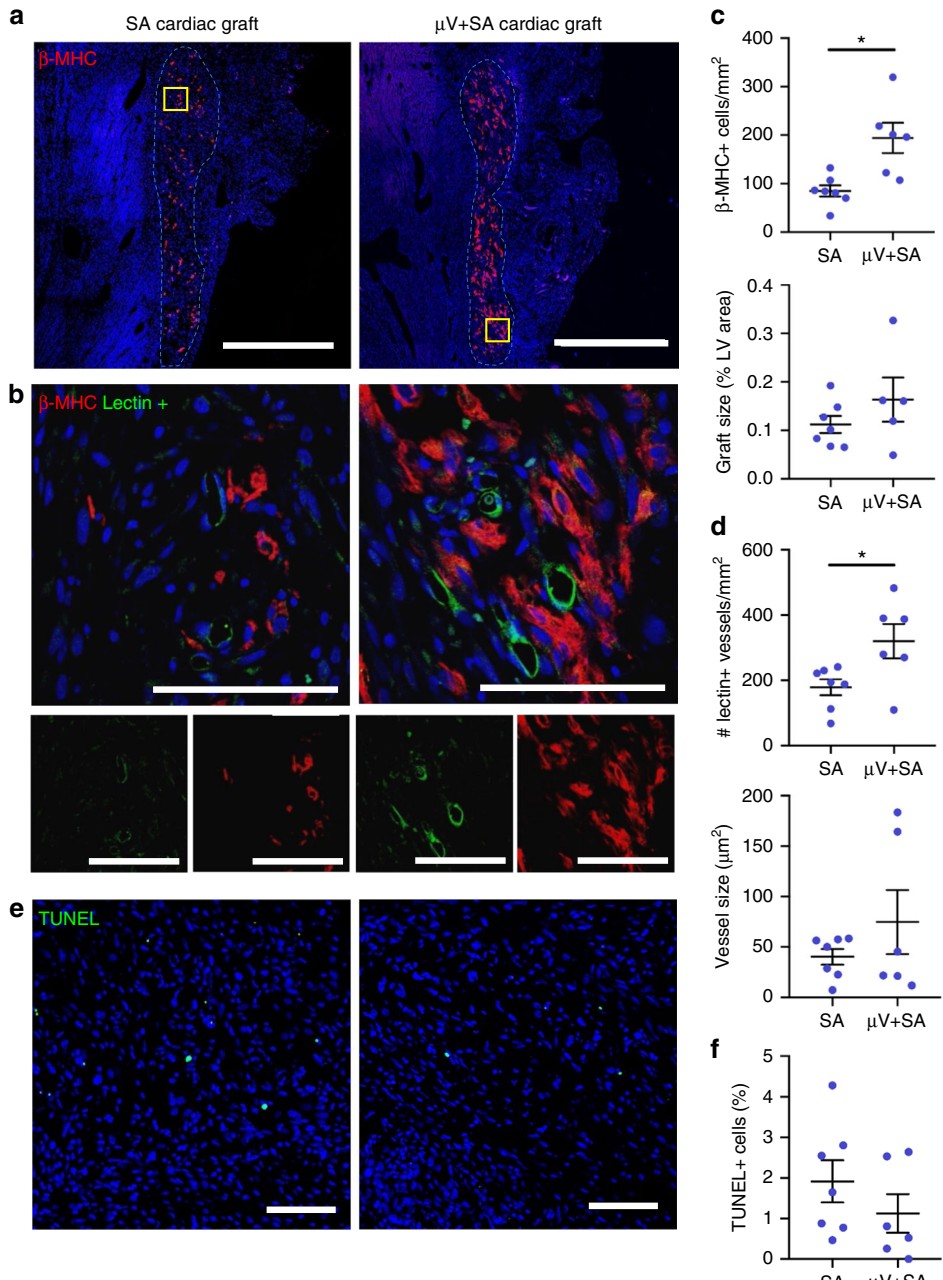

**Fig. 7** Perfusable cardiac constructs in infarcted rat heart model 5 days post-implantation **a**, **b** Immunofluorescent-stained paraffin sections of self-assembled (SA; left) and µV+SA (right) cardiac grafts: **a** Whole-graft section containing human embryonic stem cell-derived cardiomyocytes (hESC-CMs) (β-MHC, red). Gray dotted line outlines graft tissue. Scale bar, 1 mm. **b** High magnification images of boxed region in **a** (yellow box) with hESC-CMs (β-MHC, red, bottom right) and perfused vessels (Lectin+, green, bottom left). Merged, top. Scale bar, 100 µm. **c** Quantification of cardiomyocyte density (number of β-MHC+ cells per mm$^2$) and β-MHC+ graft size (% LV area). $N = 7$ and 6 biologically independent animals for SA and µV+SA, with one animal with µV+SA graft excluded from graft size calculation due to partial graft removal during tissue processing. $p = 0.015$ for density, $p > 0.05$ for graft size (two-tailed $t$ test). **d** Quantification of vessel density (number of vessels per mm$^2$) and average vessel size for all perfused vessels (Lectin+) in the cardiac grafts. $N = 7$ and 6 biologically independent animals for SA and µV+SA, respectively. $p = 0.046$ for vessel density, $p > 0.05$ for vessel size (two-tailed $t$ test). **e** Terminal deoxinucleotidyl transferase-mediated dUTP-fluorescein nick end labeling (TUNEL) assay for apoptotic cells (TUNEL, green) on paraffin sections of SA (left) and µV+SA (right) cardiac grafts. Scale bar, 100 µm. **f** Quantification of apoptotic cells (TUNEL+) as a percentage of all cells (human and host) in the graft. $N = 7$ and 6 biologically independent animals for SA and µV+SA, respectively. $p > 0.05$ (two-tailed $t$ test). Representative images for **a**, **b**, **e** from six biologically independent animals containing SA grafts and five biologically independent animals containing µV+SA grafts, with similar results. Hoechst-stained nuclei, blue. Data were collected from at least two confocal images of randomly selected regions per sample. Images analyzed by a custom lumen identifying code for vessel density and size. Error bars, mean ± SEM. *$p < 0.05$ determined using two-tailed $t$ test. β-MHC beta-myosin heavy chain

determines host vessel infiltration patterns and leads to better vascular structure and function.

Indeed, several studies have shown that prior vascular patterning in grafts promotes better overall integration[41,42], presumably as a result of topographical cues that provide guidance to the invading host blood vessels[15,43,44]. Given this extensive remodeling process, it would be interesting to determine whether the precise geometry of the vasculature affects its ability to integrate. Here we used a grid-like pattern to fabricate the µVs in order to maximize the perfusion area with multiple vascular branches. In future applications, the network structure could be generated with aligned parallel vessels or acute angle branches to better mimic coronary structure. Juhas et al. recently demonstrated that the aligned structure of muscle fibers can provide structural cues and guidance to invading host vasculature[45]. In the context of the heart, it is possible that the presence of oriented cardiomyocytes along with patterned parallel vasculature could help guide coronary infiltration to better mimic myocardial structure. In our vascularized grafts, the patterned vasculature provides support for the surrounding cardiomyocytes, suggesting the important role of vascular patterning and perfusion in tissue remodeling and host integration.

Our work demonstrates improved integration in the heart using patterned perfusable vasculature and quantitatively assessed graft perfusion dynamics and demonstrate their contribution to support cardiac grafts after implantation. This work took important steps toward addressing several of the key challenges associated with cardiovascular tissue engineering, namely, vascularization and host integration, and will have numerous implications in future cardiac tissue engineering and regenerative approaches.

## Methods

**Generation of GFP- and mTm-hESC-ECs.** All experiments used cells derived from RUES2 (Rockefeller University, NIH 0013) hESCs. In vitro µV experiments used genetically modified RUES2 hESCs that contain a dual-reporter (mTmG-2a-Puro) transgene[26]. Cells were maintained in undifferentiated colonies on Matrigel (BD Biosciences) coated plates with mTeSR-1 (Stemcell Technology) or mouse embryonic fibroblast conditioned media (MEF-CM) supplemented with 5 ng/mL human basic fibroblast growth factor (bFGF) (Peprotech)[46]. Untreated cells with the dual-reporter transgene stably express tdTomato red fluorescent protein (mTm). To generate a stable GFP stem cell line, mTmG-2a-Puro hESCs were treated with 5 µM de-salted Cre recombinase (Excellgen) for 24 h to induce recombination and excision of TdTomato followed by two 24 h doses of 5 ng/mL puromycin (InvivoGen) to purify the recombined population. GFP-expressing cells were expanded and analyzed for GFP purity using flow cytometry on unfixed and unstained cells. hPSCs were differentiated into ECs through cardiac mesoderm using high Activin A and low BMP4[16,17]. Briefly, single-cell suspensions of non-colored or mTm- and GFP-expressing hESCs were obtained using enzymatic dissociation and re-plated on Matrigel at a density of 105K cells/cm² in mTeSR-1 or MEF-CM+5 ng/mL bFGF and 1 µM CHIR-99021 (Cayman Chemical). When the plated cells reached about 80% confluent monolayer, they were fed with day 0 induction media (100 ng/mL Activin A (R&D) in RPMI (Gibco, with L-glutamine) +B27 supplement (Thermo Fisher, without insulin) and 1× Matrigel. Eighteen hours post-induction, the cells were fed with 5 ng/mL BMP4 (R&D) and 1 µM CHIR-99021 in RPMI+B27 (without insulin). Twenty-four hours later, the media was changed to endothelial induction media: Stempro34 (Invitrogen) containing 200 ng/mL VEGF (Peprotech), 5 ng/mL bFGF, 10 ng/mL BMP4, 4 × 10⁻⁴ M monothioglycerol (Sigma), 50 µg/mL Ascorbic Acid (Sigma), 2 nM L-Glutamine (Invitrogen), and pen–strep (Invitrogen). The cells remained in endothelial induction conditions for 72 h, with no media changes. At the end of 72 h (differentiation day 5), endothelial progenitor cells were harvested for re-plating and flow cytometric analysis. The cultures were fed with fresh Stempro (with L-glutamine and pen–strep) 1–2 h prior to re-plating to improve survival and attachment. Cells were then enzymatically harvested to obtain a single-cell suspension and re-plated in 0.2% gelatin-coated tissue culture flasks with endothelial growth media (EGM, Lonza) containing 20 ng/mL VEGF (Peprotech), 20 ng/mL bFGF, and 1 µM CHIR-99021. An aliquot of day 5 cells were stained with CD34–allophycocyanin FACs antibody (BD 555824, 1:4) and analyzed with flow cytometry to determine differentiation efficiency of endothelial progenitors. Re-plated cells were maintained in EGM with CHIR-99021, bFGF, and VEGF (EGM+ Factors). In all experiments, hESC-ECs were harvested at differentiation days 10–14 and then used to generate engineered µVs. Aliquots of differentiated cells were stained with CD31–phycoerythrin (PE) FACs antibody (BD 555446, 1:4) and analyzed with flow cytometry to determine endothelial purity. All staining for flow cytometry was done on live cells. Aliquots were washed in phosphate-buffered saline (PBS) containing 5% fetal bovine serum (FBS), resuspended in Dulbecco's modified Eagle's medium (DMEM) media (Corning) with conjugated antibody, and kept in the dark on ice for 45 min. Stained cells were washed and analyzed immediately with FACs Canto II cell analysis instrument and FlowJo Software.

**Generation of hESC-CMs.** Noncolored RUES2 hESCs were differentiated into hESC-CMs using modulation of Wnt signaling[47]. Briefly, undifferentiated cells were maintained as described above in mTeSR-1. Single-cell suspensions of non-colored hESCs were dissociated and replated at a density of 40K cells/mm² in a Matrigel-coated 12-well plate in mTeSR-1. After 24 h, cells were supplemented with 1 µM CHIR-99021. At 48 h after replating, cells were fed with day 0 induction media of 3 µM CHIR-99021 in RPMI containing 500 µg/mL bovine serum albumin (BSA; Sigma) and 213 µg/mL ascorbic acid (RBA). On day 2, media was changed to 2 µM WNT-C59 (Selleck) in RBA media. On day 4, media was changed to RBA without supplementation. Beginning on day 6, cells were fed with RPMI +B27 supplement (Thermo Fisher, with insulin) on every other day until harvest. Initial beating was observed on days 6–8. In all experiments, cardiomyocytes were enzymatically harvested into a single-cell suspension at differentiation days 29–31 and then used to generate cardiac constructs. Aliquots of differentiated cells were stained with mouse anti-cTnT (ThermoScientific MS-295, 1:100) and goat anti-mouse IgG-PE (Jackson Cat 115–116–072, 1:200) for 30 min each in PBS containing 0.75% saponin and 5% FBS (Gibco). Stained cells were analyzed by flow cytometry with FACs Canto II cell analysis instrument and FlowJo Software to determine cardiomyocyte purity. For all cardiac constructs, hESC-CMs had cardiomyocyte purity >70% (Supplementary Figure 11A-B).

**Fabrication and culture of engineered µVs.** Engineered µVs were generated using injection molding and soft lithography techniques[28,29]. Briefly, for EC only µV constructs, 15 mg/mL stock collagen (dissolved in 0.1% acetic acid[28]) was neutralized and diluted to 6 mg/mL on ice with 1 M NaOH (20 mM final), 10× M199 (Sigma), and EGM. For constructs containing cardiomyocytes, 15 mg/mL stock collagen was diluted to 6 mg/mL on ice with 1 M NaOH (10 mM final), 10× RPMI (Sigma, 1× final), 30x Matrigel (1× final), 100× Hepes (Invitrogen, 1× final), and RPMI+B27 supplement[11]. For experiments containing hESC-ECs in the bulk collagen, the cells were added after the collagen was neutralized at a density of 3 × 10⁶/mL, and the volume of EGM was adjusted accordingly to compensate for the additional volume of cells. For cardiac constructs, cells were added to collagen mixture at densities of 12 × 10⁶/mL hESC-CMs, 3 × 10⁶/mL hESC-ECs, and 3 × 10⁶/mL human bone marrow-derived stromal cells (HS-27a). HS-27a (courtesy of Beverly Torok-Storb laboratory at Fred Hutchinson Cancer Research Center) were maintained on 0.2% gelatin-coated flasks in DMEM GlutaMAX (Thermo Fisher) containing 20% FBS and 1× pen-strep.

Specially designed devices that contain holes for collagen injection, inlet, and outlet were used during injection molding, assembly, and µV culture. The microchannels were created by molding liquid 6 mg/mL collagen (acellular (µV only) or hESC-EC containing (µV+SA)) around a microfabricated polydimethylsiloxane (PDMS) stamp that contains vessel network geometry. For in vitro experiments, 3 × 3 geometry with 100 µm feature height was used (Supplementary Figure 3A). After gelation, the channel was enclosed with a separately generated flat layer of collagen to create embedded microfluidic channels surrounded by native collagen matrix on all sides. Day 10–14 noncolored or mTm-hESC-ECs were seeded in the channel by perfusing at a density of 10 × 10⁶/mL. The cells were allowed to circumferentially attach overnight in static conditions, followed by flow through the lumen to remove the non-attached cells in the lumen and reservoir. Constructs were then cultured under gravity-driven flow for 4 or 7 days with EGM+ factors media. Gravity-driven flow is generated by replacing the media at the device inlet every 12 h for the duration of the culture period. These culture conditions generate sheer stresses on the order of 0–10 dyne/cm² with a time average of 0.1 dyne/cm² and the highest sheer application immediately after media replacement[28]. For perfusable cardiac constructs, housing devices contained pieces that were removed after collagen gelation to allow for additional media access. Cardiac constructs were submerged in 25% EGM+ factors and 75% RPMI +B27 supplement.

**Analysis of in situ µV immunofluorescence and bead perfusion.** To assess the cell death in µVs, vessel lumen was perfused with 10 µM ethidium homodimer 1 and Hoechst for 1 h followed by 3 × 5 min PBS washes, then imaged on a Nikon A1R confocal microscope for z-stack images before fixation. ImageJ particle analysis for red channels was within the specified distance ranges from the µVs: [0–300 µm], [300–600 µm], and [600–900 µm]. In other conditions, engineered vessels were perfusion-fixed after 4 or 7 days in culture with 3.7% formaldehyde (Sigma) for 20 min followed by 3 × 20 min PBS washes. Samples were blocked in 2% BSA (Sigma) and 0.5% Triton X-100 (Sigma) in PBS for 1h followed by an overnight incubation with the following primary antibodies at 4 ℃: rabbit pAb to hCD31 (abcam 28364, 1:25), mouse pAb to VE-Cadherin (abcam 7047, 1:50), rabbit pAb to DsRed (abcam 16667, 1:100), and goat pAb to GFP (abcam 5450,

1:400). One-hour incubation with the following secondary antibodies, conjugated primary antibodies, and nuclei counterstains was performed at room temperature: sheep pAb to Von Willebrand Factor (abcam 8822, 1:100), donkey anti-goat Alexa Fluor 488 (Invitrogen A11055, 1:100), donkey anti-mouse Alexa Fluor 647 (Invitrogen A31571, 1:100), donkey anti-rabbit Alexa Fluor 594 (Invitrogen A21207, 1:100), and Hoechst 33342 (Sigma, 1:250). 3D z-stack images were acquired on a Nikon A1R confocal microscope with all image post-processing and quantification done with the ImageJ software. GFP-hESC-EC density and lumen size were quantified in μV+SA constructs by applying a threshold to maximum intensity projections (MIPs) of the GFP channel followed by ImageJ particle analysis (particles <20 μm$^2$ in size excluded to account for background) within the specified distance ranges from the μVs: [0–300 μm], [300–600 μm], and [600–900 μm]. Endothelial sprouts were quantified with either DsRed-stained μVs (μV+SA) or CD31-stained μVs (μV only) to determine sprouting from the patterned μV since GFP-hESC-ECs also express CD31. Sprouts were manually counted and their lengths and diameter measured using 3D orthogonal views and z-stack images in ImageJ. The number of sprouts was normalized to the surface area of the endothelium. After all confocal images were collected, vessels were perfused with 50 μL of 1.0-μm diameter red fluorescent beads (Thermo Fisher F13083, 1 × 10$^{10}$ beads/mL, 580/605) diluted 1:30 in PBS. Brightfield and fluorescent time series were collected with no delay at ×10 magnification on a Nikon high-resolution wide-field microscope. Bead velocity was quantified with manual particle tracking in both sprouts and the patterned microchannel (D > 100 μm). Perfusion area was quantified by measuring the total area of perfused beads in large images that were manually stitched together with the ImageJ software from smaller ×10. For each vessel, the stitched image encompassed a field of view twice the area of the original pattern boundary.

**Whole-blood perfusion and analysis of platelet accumulation**. Fresh blood (in 0.129 M sodium citrate) was obtained through Puget Sound Blood Center under University of Washington Institutional Review Board-approved protocols from consenting healthy donors. Platelets in the platelet-rich plasma (PRP) were labeled with fluorescein isothiocyanate-conjugated CD41a antibody (BD 555466, final concentration of 2.5 μg/mL) for 30 min at room temperature. The labeled PRP were reconstituted with red blood cells and buffy coat at the original ratio to produce reconstituted whole blood, which was perfused through the live μVs that had first been washed with PBS. For activated μVs, 50 ng/mL PMA or 1 μg/mL IL-1β were perfused through the vessel lumen for 30 min before PBS wash. The blood flow was set for 15–30 min so that the wall shear stresses are between 10 and 30 dyne/cm$^2$. All blood perfusion experiments were done in mTm-hESC-EC seeded μVs that were cultured for 4 days. Brightfield and fluorescent no delay time series were collected at ×10 magnification on a Nikon high-resolution wide-field microscope to visualize red blood cell movement and platelet accumulation of the vessel walls. At the end of blood perfusion, the vessels were washed with PBS for three times to remove the excessive blood-cell suspensions and fixed in situ using 3.7% formaldehyde. The vessels were then washed with PBS three times and immunohistochemically stained for CD31 and VWF[28] and imaged on a Nikon A1R confocal microscope to obtain image stacks along z direction. Using the ImageJ software, the projections of image stacks were made and platelet adhesion was quantified by measuring the area of CD41a+ fluorescent signal and normalizing to vessel wall surface area.

**RNA isolation and RNAseq data analysis**. Total RNA from three types of constructs was purified using the RNAeasy Mini Kit (Qiagen) and residual DNA was removed by on-column DNase digestion. RNA quality was assessed with the Agilent RNA 6000 Nano Kit using the Agilent 2100 Bioanalyzer (Agilent Technologies). Only samples with RNA integrity number >8 were kept for further analysis. RNA sequencing was performed on poly-A-enriched samples using Illumina TruSeq.

RNAseq samples were aligned to hg38 using TopHat v2.1.0[48]. Gene-level read counts were generated with htseq-count v0.6.1p1[49] using the intersection-strict overlapping mode. Genes with >1 reads per million in at least 2 samples were kept for further analysis. The GLM method in edgeR v3.18.1 was used for differential expression analysis[50]. prcomp function from R was used for PCA. Genes with fold change >1.5 and FDR < 0.01 were considered differentially expressed. Hallmarks gene set[51] was used for pathway enrichment analysis. Hypergeometric test was used to test the significance of overlap between upregulated/downregulated genes and a pathway.

**Rat I/R and vascular implantation surgeries**. All animal procedures in this study were approved by the University of Washington Institutional Animal Care and Use Committee (IACUC, protocol #2225–04) and performed in accordance with US NIH Policy on Humane Care and Use of Laboratory Animals. Male athymic nude Sprague-Dawley rats (approximately 250–300 g, 8 weeks of age) underwent two thoracotomy surgeries for I/R injury and implantation of vascular grafts (4 days after I/R). Subcutaneous administration of sustained release buprenorphine (1 mg/kg) provided analgesia for at least 2 days after all surgeries. The animals were closely monitored for 48 h to provide post-operative care to ensure and maintain animal health and comfort as outlined in the IACUC protocol. Cyclosporine A

treatment (5 mg/kg) to prevent mitochondrial permeability transition pore opening in the grafts was administered subcutaneously for a total of 7 days beginning the day before implant surgery. For the I/R surgery, the rats were anesthetized with an intraperitoneal (IP) injection of 68.2 mg/kg ketamine and 4.4 mg/kg xylazine. A second dose of full-strength ketamine/xylazine followed by additional ketamine boosts (20 mg/kg, administered as needed) were used to maintain a surgical plane of anesthesia. During the procedure, the rats were intubated, mechanically ventilated, and maintained on a water-circulating head pad. Core body temperature was monitored at regular intervals and maintained at 37 °C. To induce myocardial infarction, the heart was exposed, and the left anterior descending coronary artery was ligated and occluded for 60 min, followed by reperfusion and aseptic chest closure. For construct implantation, patterned vascular constructs were made with mTm-hESC-ECs in both the bulk collagen matrix (3 × 10$^6$/mL) and seeded in the microchannel (μV+SA) as described above. Control non-patterned constructs (SA) were made by mTm-hESC-EC suspension in 6 mg/mL collagen (3 × 10$^6$/mL) followed by gelation in an 8 mm diameter × 1 mm height PDMS well. The number of cells attached in the luminal surface of microchannels of μV+SA construct is very close to the number of cells in SA only conditions taking the space of the volume of microchannels. Therefore, both SA only and μV+SA constructs have similar cell number (approximately 150,000 cells per construct with 10,000 cell difference). Cardiac constructs of both μV+SA and SA only conditions were made with hESC-CMs (12 × 10$^6$/mL), noncolored hESC-ECs (3 × 10$^6$/mL), and HS-27a (3 × 10$^6$/mL) in the bulk in 6 mg/mL collagen. Perfusable cardiac constructs were seeded with noncolored hESC-ECs along the luminal walls and cultured under gravity-driven flow with EGM+GFs while media of 25% EGM and 75% RPMI+B27 supplement were placed on top of cardiac construct throughout culture. Both types of constructs were cultured for 4 days in vitro to allow initial remodeling and anastomosis of angiogenic sprouts from the channels to de novo tubules from bulk-seeded ECs. One day prior to surgery, cardiac constructs were heat shocked at 42 °C for 1 h. On the day of implant surgery, the rats were anesthetized via inhalation of isoflurane at 5%. The rats were intubated and mechanically ventilated with continued isoflurane supplementation (3%) to maintain a surgical plane of anesthesia. Core body temperature was monitored at regular intervals and maintained with a water-circulating head pad at 37 °C. Immediately prior to implantation, the constructs were carefully removed from their housing devices and placed in warm media bath for vascular constructs or pro-survival media on ice for cardiac constructs. Pro-survival media consisted of the following medias at a 1:1:2 ratio: EGM+ factors, RPMI+B27 supplement, and pro-survival cocktail containing Matrigel (30× final), 200 nM cyclosporine (Novartis), 100 μM ZVAD (benzyloxycarbonyl-Val-Ala-Asp (O-methyl)-fluoromethyl ketone, Calbiochem), 50 nM Bcl-X$_L$ BH4 (cell-permeant TAT peptide, Calbiochem), 100 ng/mL IGF-1 (Peprotech), and 50 μM pinacidil (Sigma) in RPMI[46]. For μV+SA constructs, an 8-mm diameter biopsy punch was used to remove the 1-mm-thick patterned region of the collagen gel from excess surrounding collagen. Additional 2 mm biopsy punches were created at the inlet and outlet to create structural conduits from the network to the underside of the construct. The constructs were implanted on the epicardial surface of the left ventricle with 8–0 surgical suture at 3–4 points of contact. Once the implant was secured, excess blood was dabbed clean followed by aseptic chest closure.

Sample sizes were determined based on pilot studies without rigorous statistical analysis. Experimental groups were assigned randomly in terms of the rat order and construct types for implantation by two independent observers.

**Tissue harvest and retrograde perfusion fixation**. At their experimental endpoint, the rats were euthanized with a chemical overdose of pentobarbital/phenytoin solution (Beuthanasia; 1.5 mL IP injection). Once the animals achieved deep anesthesia but while the heart was still beating, the chest was opened and 50 U Heparin was intravenously infused via the inferior vena cava and allowed to circulate for 1–2 min to prevent thrombosis in the coronary vessels. Intravenous infusion with supersaturated potassium chloride (KCl) was then used to arrest the heart in diastole followed immediately with excision of the heart. The aorta was cannulated followed by retrograde perfusion with a vasodilator buffer (PBS containing 4 mg/L Papaverin and 1 g/L adenosine) followed by 4% paraformaldehyde perfusion for 10 min. Perfusion pressure was maintained at ~100 mm Hg. After perfusion fixation, the hearts were transferred to fresh fixative overnight at 4 °C. After overnight fixation, the cannulated hearts were transferred to PBS buffer and then transferred on ice for OMAG imaging.

**OMAG assessment of vascular flow in grafts and normal healthy heart**. To image vascular flow, the hearts were retrograde perfused with 10% Intralipid (Sigma)[25] and imaged with OCT-based technology[52]. During imaging, the hearts were placed on a custom-built imaging platform with rotational control and secured cannula to prevent tissue movement[20]. Perfusion pressure was maintained at a diastolic pressure of 90 mm Hg throughout image acquisition. Two scanning protocols that covered the same 1.5 × 1.5 mm$^2$ field of view were used to acquire data for both OMAG and OMAG-V. Both protocols shared the same system set-up with conventional fiber-based spectral domain OCT, which has been described in detail in a previous publication[53]. Briefly, this section will describe the specific parameters that determined the system performance for imaging rat coronary vessels ex vivo. The light source was a super-luminescent diode (LS2000B, Thorlabs Inc.) with a spectral bandwidth of 110 nm at 3 dB and operated at a center

wavelength of 1340 nm. A ×10 telecentric objective lens was adopted to focus the beam spot onto the heart sample with an incident light power of ~1.9 mW. The axial resolution was ~7 μm and the lateral resolution was ~ 7 μm in air. The maximal imaging speed of the system reached 92,000 A-lines per second, and the corresponding system sensitivity was ~100 dB in focus positioned at ~600 μm below the zero delay line. The system ranging depth was measured to be 3.5 mm in air.

For OMAG, raster beam scanning was performed to capture 250 A-lines within each B-frame (2D cross-section), 4000 B-frames for each C-scan (3D volume), and 16 frame repetitions at each of the 250 A-line locations to achieve high vasculature contrast. A single 3D dataset was obtained in 15 s by using a frame rate of 280 frames/s. MIPs of the volumetric vasculature were used to calculate vessel density as a percentage of the imaging field of view.

For OMAG-V, 200 B-frames were captured in each C-scan. Within each frame, 10,000 A-lines were obtained that were comprised of 50 repeated A-lines at 200 positions. By utilizing a defined system speed of 20,000 A-lines per second, the 3D velocimetry data were acquired in 2.5 min. An inter-frame ultrahigh-sensitive OMAG algorithm[53] was utilized to extract 3D vasculatures from heart tissue structure images. By MIP of the volumetric vasculature, morphological information of the microvascular network can be visualized from the top view. In order to quantitatively analyze the flow velocity in capillaries, Eigen decomposition statistical analysis was applied to the repeated A-lines[35,53]. In brief, the repeated A-line ensembles were first stacked into a covariance matrix, and then this matrix was represented in terms of its eigenvalues and eigenvectors through diagonal factorization. Therefore, the frequency of flowing intralipid particles can be calculated by the first lag-one autocorrelation of the eigenvectors[35]. Finally, the velocity of flow in vessels with diameters between 20 and 40 μm was assessed according to its linear correlation with the measured frequency[35]. Graft perfusion rates for the imaging field of view were calculated by multiplying the velocities with the perfused area and normalizing to the perfusion rates of the corresponding healthy region. All analysis of animal experiments was performed by a blinded observer without knowledge of the experimental group.

**Histological assessment of grafts and lumen quantification**. Following OMAG imaging, the hearts were perfused with PBS to wash out remaining intralipid solution followed by perfusion for 10 min with a 1:1 mixture of fluorescein-labeled or unconjugated GSL I (Vector Labs, 8 μg/mL) and rhodamine-labeled UEA 1 (Vector Labs, 8 μg/mL) in order to label the ECs of perfused rat (GSL I-bound) and human (UEA 1-bound) endothelium. The hearts were flushed with PBS and then sliced into 2-mm-thick sections from the apex for paraffin processing and embedding[31,46]. Four-μm sections were cut and stained for picrosirius red/fast green (to assess infarct size) or subjected to immunohistochemistry[31,18]. Briefly, slides were de-paraffinized and rehydrated followed by enzymatic antigen retrieval (EAR) with proteinase k (Roche, 15 μg/mL in 10 mM Tris/HCl) at 37 °C for 20 min or heat-induced epitope retrieval (HIER) for 20 min in boiling Tris/EDTA buffer (pH 9.0) or citrate buffer (pH 6.0). The samples were blocked and permeabilized with natural donkey serum (Jackson, 10%) and 0.5% Triton X-100 followed by overnight incubation with primary antibodies: rabbit pAb to DsRed (Clontech 632496, 1:75, EAR) to detect TdTomato reporter in mTm-hESC-ECs, rabbit pAb to hCD31 (abcam 28364, 1:50, HIER Tris/EDTA) to detect hESC-ECs, mouse mAb to smooth muscle actin (abcam 7817, 1:100), goat Ab to GSL I (Vector AS-2104, 1:200), mouse mAb to rhodamine (abcam 9093, 1:150), mouse Ab to rat CD68 (Serotec MCA341/GA, 1:100, EAR) to detect macrophages, mouse mAb to sarcomeric α-actinin (abcam 9465, 1:50, EAR), and mouse mAb to human β-MHC (hybridoma supernatant, ATCC #CRL-2046, full strength, HIER citrate). For immunofluorescent-based stains, Alexa Fluor-conjugated secondary antibodies (Invitrogen) were used along with Hoechst (Sigma, 1:250) nuclei counterstain. Apoptotic cells were identified using a Click-iT Plus TUNEL Assay with Alexa Fluor 594 dye (Thermo Fisher). For brightfield peroxidase detection of macrophages or human β-MHC (graft size), a biotintylated goat anti mouse secondary antibody (Jackson, 1:500) was used in conjunction with the Vectastain Avidin/Biotin Complex (ABC) Kit (Vector labs) and 3,3'-diaminobenzidine (Sigma) followed by routine hematoxylin counterstain to detect nuclei. The density and size of perfused lumens in vascular grafts were quantitated with custom Matlab code to analyze ×20 confocal microscopic images of slides stained for DsRed, GSL I, or UEA I. Please note that, while the lectins GSL I and UEA I are species specific in their endothelial-binding affinity, the antibodies against them were not. Thus anti-GSL I stains were used to evaluate perfused lumens of both species, while stains for anti-rhodamine were used to detect perfused human vessels. The graft size was determined by total β-MHC+ area relative to total left ventricle area on a 4-μm section containing 4–5 heart sections. The number of cardiomyocytes per area (cardiomyocyte density) was quantified by manual cell counting of β-MHC+ cells in ×20 confocal microscopic images. Overall and cardiomyocyte-specific viability within the graft was determined by total nuclei or sarcomeric α-actinin+ cell colocalization with TUNEL+ marker determined by particle analysis of ×40 confocal microscopic images on ImageJ. All analysis of animal experiments was performed by a blinded observer without knowledge of the experimental group.

**Statistical analysis**. Unless otherwise noted, single variable analysis with two-tailed *t* test assuming unequal variance was used to determine statistical significance between two samples. All results are presented as mean ± standard error and assumed to be distributed approximately normally. For in vitro μVs, the sample number represents the number of constructs analyzed unless otherwise noted. For in vivo experiments, the sample number per group represents the number of animals. Significance was defined as *$p < 0.05$ and **$p < 0.01$ for all results.

**Code availability statement**. All codes used in this study are available from the corresponding authors upon request, except the custom codes used for OMAG and OMAG-V acquisition and analysis are available from Dr. Ruikang Wang (wangrk@uw.edu) upon request. The custom Matlab code for quantifying density and size of perfused lumens in vascular grafts has been deposited in the GitHub database under accession code [https://zenodo.org/record/2434926]. Contact the corresponding authors for more information.

**Reporting summary**. Further information on experimental design is available in the Nature Research Reporting Summary linked to this article.

## Data availability
Additional information from the study is available from the corresponding authors upon request. RNA-seq data have been deposited in the Gene Expression Omnibus database under accession code GSE124314.

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

## Acknowledgements

We acknowledge the Lynn and Mike Garvey Imaging Laboratory in the Institute of Stem Cell and Regenerative Medicine, the Nanotech User Facility, and Flow Cytometry Facility, all at the University of Washington. We thank Ms. Jun Xue and Mr. Daniel Lih for their help in endothelial cell and cardiomyocyte differentiation; and Dr. Lil Pabon and Dr. Hans Reinecke for helpful discussions along the course of these experiments. We acknowledge the financial support of National Institute of Health grants R01HL141570 (to Y.Z. and C.E.M.), DP2DK102258 (to Y.Z.), P01HL094374, R01HL128362, and P01GM081619 (to C.E.M.), R01EY024158 and R01HL093140 (to R.W.), T32HL7312 (to M.A.R., training grant), and T32EB1650 (to N.Z., training grant). We also acknowledge the support of the Foundation Leducq Transatlantic Network of Excellence (to C.E.M.).

## Author contributions

C.E.M. and Y.Z. conceived the project; M.A.R., N.Z., C.E.M., and Y.Z. designed the experiments. M.A.R., N.Z., A.M., and Y.Z. performed both in vitro and in vivo experiments; M.A.R. and N.Z. analyzed the data; M.A.R., W.Q., and W.W. performed OMAG imaging and analysis; Y.W. and Y.Z. performed RNAseq analysis; M.A.R., N.Z., C.E.M., and Y.Z. interpreted the data and wrote the manuscript. R.K.W. oversaw the OMAG experiments; C.E.M. and Y.Z. oversaw all phases of the project and obtained research funding. All authors interpreted the data, edited, and approved the manuscript.
