## [Peer Review File · Nature Communications]

Reviewers' Comments:

Reviewer #1:

Remarks to the Author:

Redd et al. present data on micro-fabrication of perfusable channels and their use in tissue engineering. The key result of the study is that engineered micro-channels are blood perfusable. A surprising finding was the loss of the preformed channels and the main contribution by endogenous vascular cells to graft perfusion. An innovative in vitro OCT-imaging technique is described for quantification of perfusion in situ.

Major concerns:

The increased vessel density and perfusion in the $\mu V+SA$ constructs may be the result of higher EC loading. If I get the methods section right, there was bulk seeding of 3 million ECs in the SA constructs and additional perfusion seeding of the micro-channels with 10 million cells in the $\mu V+SA$ group.

EC survival after bulk seeding appeared to be a function of distance from the fabricated micro-channels. This is surprising given a construct thickness of 100 μm for the in vitro studies. Life-dead stains or time-course assessment of apoptosis would help to better understand the differences in EC survival

Several groups have demonstrated vascularization of tissue grafts. The study would be markedly improved if enhanced survival of co-seeded cardiomyocytes could be demonstrated.

The true benefit of the micro-fabrication approach is not clear if for the most part vascularization is from endogenous cell sources.

The claim of having fabricated highly vascularized constructs should be toned down. Vascular density, mainly due to capillary ingrowth, remains rather low in vivo. As to the in vitro data, it is important to differentiate between cell-seeded micro-channels and bona fide vessels or capillaries. For the most part it is the channels that contribute to perfusion and not the EC sprouts.

Minor concerns:

The title should be revised to not overstate the key finding, which is the engineering of a blood perfusable microchannel system and its application to enhance graft vascularization by endogenous ECs. Evidence for outcome relevant anastomoses is not provided.

Figure legends need to be carefully revised. The provided information is in some legends insufficient to fully understand the panels, e.g., Fig 1a and SFig 2a, and sometimes lacking completely, e.g., Fig 2G-H. In Fig. 2B, arrows cannot be distinguished well.

Data on vessel density in the infarct area and a comparison to graft vessel density would be informative.

Please, provide control lectin labelling data to confirm species specificity.

Fig 4 panels C and D: a clarification is needed as to the differences in human vessel count

Reviewer #2:

Remarks to the Author:

In this study, the authors engineered perfusable microvascular constructs, wherein human

embryonic stem cell-derived endothelial cells (hESC-ECs) were seeded both into patterned channels formed within a collagen matrix and the surrounding interstitial space. They compared these perfusable constructs in vitro, and to those containing only bulk-seeded endothelial cells for promotion of vascular anastomosis and host integration with infarcted hearts in vivo. The hESC-ECs lining the luminal walls readily sprouted and anastomosed with de novo-formed endothelial tubes in the bulk matrix in vitro. Flow through the patterned microvessels in vitro promoted connectivity and perfusion in anastomotic regions and enhanced the vascular density and average lumen size in the matrix. These perfusable constructs also exhibited nonthrombotic nature when perfused with human whole blood. When implanted on infarcted rat hearts, the perfusable microvessel grafts integrated with coronary vasculature to a greater degree than non-perfusable self-assembled vascular constructs at 5 days post implantation. Optical microangiography imaging in intact whole hearts revealed that perfusable grafts had 6-fold greater vascular density, 2.5-fold higher vascular velocities and >20-fold higher volumetric perfusion rates. Thus, the authors conclude that pre-patterned vascular networks enhance vascular anastomoses in vitro and accelerate coronary perfusion by the host after implantation in vivo.

This is an extension of work conducted by this group for some time. The work presented is not conceptually novel as this group and others (Sasagawa T et al. J Tiss Engineer Regen Med 2016; Kang K-T et al. Blood 2011) have shown that in vitro patterned bioengineered vascular constructs can be shown to generate vessels in vivo when implanted into various tissues. This present work is an incremental advance to show that patches of prevascularized patterned collagen implants with bulk seeded hESC-EC demonstrate greater anastomosis with vessels in host injured tissues. The speed with which these implanted pre-patterned vascular structures connected with host vasculature is similar to prior reports of 3-5 days. Unfortunately, there are no mechanistic insights into how this type of implant is working to enhance vascularization of the ischemic tissue, and surprisingly despite improved blood flow, there is similar levels of ischemic heart damage and inflammatory cell infiltrate. This suggests that the accelerated perfusion of the tissue may not enhance the recovery of the damaged myocardium.

Reviewer #3:

Remarks to the Author:

The authors demonstrated that providing pre-patterned microvascular grafts, as opposed to tubes formed de novo by culturing suspended single cells, results in increased perfusion when grafted in the injured rat myocardium. They show that the perfused grafts are lined with the originally seeded hESC-ECs and host endothelial cells, the latter of which predominate. The novelty is that these data is that the authors are the first to demonstrate that micro-patterned vessels may contribute to an increase in perfusion in vivo, which is can be detected in a quantitative manner using optical microangiography. This research has the potential to provide valuable evidence that patterned vessels increase perfusion in vivo, which is a crucial first step in clinically restoring blood flow to ischemic regions of the heart.

The manuscript also includes data describing the organization of hESC-ECs in the pre-patterned device, and a lack of a dramatic effect of the addition of single cells within the surrounding matrix. Importantly, it is shown that the pre-patterned grafts do not cause coagulation. Overall, the results are a step towards improving the performance of grafted tissue into injured myocardium and detection of perfusion in these grafts. However, several of the conclusions are not fully supported by the data as shown and the rationale for some of the experiments is unclear. Please see specifics below.

Major Comments:

1. The authors need to more specifically state what they are testing in Figure 1. What is their hypothesis and what conclusions are they drawing from the data. The purpose of showing a comparison between SA alone and $\mu V+SA$ is not clear since there are no significant differences between the two conditions in E. Also, the colors used in G are misleading because the same colors

are used for different conditions in E.

2. Along those same lines, there is little continuity between the in vitro and in vivo work. In vitro, the authors show a comparison between the micro-vessels (μV) alone and the micro-vessels plus self-assembled lumens ($\mu V+SA$), but in vivo the authors are comparing SA alone and $\mu V+SA$.

Please clearly state the rationale/hypothesis for these experiments? Also, it appears that the μV graft alone might be sufficient to induce perfusion into the graft since SA alone has no flow in vivo? The authors need to include the μV alone condition so that we can make a conclusion about the necessity of these two cell types for the observed effect.

3. What is the evidence that the cells are endocardial-like. There needs to be endocardial marker expression shown for the cells, not just those of endothelial cells, if this claim is to be made.

4. Figure 1B and C are said to show angiogenic activity. There are no parameters of angiogenesis shown in these panels. These same Figures are also said to show lumens, which are not shown in the micrographs.

5. For video S1, it is not obvious that these are "highly integrated microvascular networks". Yes, the green cells are intermixing with the red, but there needs to be more evidence of a network to make this conclusion. What do the authors mean by network? It usually describes a network of perfused interconnect blood vessels. Perhaps show 3D views of the confocal image and have to movie rotate it instead of scrolling through?

6. The authors conclude that: "these data suggest that the localized vascular remodeling and anastomotic events decrease the vascular resistance allowing for more efficient perfusion through endothelial sprouts". More evidence needs to be provided to make this conclusion, including a measurement or modeling of resistance in the networks.

7. In Figure 1D, evidence for a tube like vessel anastomosing with the patterned tube needs to be stronger.

8. There is no evidence of rapid anastomosis in vivo. This makes the title of the manuscript misleading because it implies that there is rapid anastomosis in vivo as well. Experiments need to be performed to provide support that rapid anastomosis plays a role in the increased perfusion they observe in vivo.

9. The thrombogenic experiments need positive (activated ECs?) and negative (non-adherent surface only?) controls. There is some adhesion. Are these physiologically relevant levels?

10. The authors need to explore what aspect of the vascular graft contributes to the increased perfusion in vivo. Most of the perfused vessels are from the host – is that solely responsible for the total increase in perfusion observed. Why are the SA cells included then? Again, looking at the μV graft alone would help.

11. Additional experiments showing improved outcome (i.e. increased ejection fraction or cardiac function) would benefit the manuscript.

Minor Comments:

1. I could not find how the cell density was normalized between the $\mu V-SA$ and SA constructs – is the effects seen only an effect of differences in cell density? Clarification is needed to ensure that the total number of cells is the same in $\mu V-SA$, μV , and SA constructs.

2. The authors should provide evidence of why they are concluding that cell death is responsible for decreased EC density 600 μm away from the patterned vessel wall in vitro.

3. Could the authors provide the figure/data for the statement: "10-fold difference between the total number of perfused vessels and the number of perfused human vessels"?

4. Typo – "(0.72 mm/s in in 20-30 ..."

5. No error bars in figure 2D and supplementary figure 4A

6. One of the images should be labeled "XY" in figure 2H, not both "XZ"

7. The caption for figure 1A and 1D are confusing because they have subsections (i-iv), but the caption doesn't explain what the differences are. It is unclear from the caption that they are related (or not related). Also consider using different numbering for 1C since it is not related to 1A and 1D (i.e. a,b,c,d). A similar issue with crossover in numbering comes up in figure 2B (since it's not related to 2A).

8. The confocal image for figure 1F seems to be tilted. Is there specific reason for this? Otherwise, just show the XY plane. Also, it is confusing where the interstitial collagen is – is it relevant to the figure?

9. Supplementary figure 3A-C should have the markers labeled on the figure itself even if the names are in the caption. In addition, adding the distance labels to the figure would make it clearer.

Response to Reviewers

Reviewer #1

We thank the reviewer for the recognition of the innovative approaches and interesting findings in our work, and helpful suggestions for improving the manuscript. In response we have now included additional data and expanded the discussion, which we hope alleviates the reviewer's concerns and improves the clarity and quality of our manuscript.

Major concerns:

The increased vessel density and perfusion in the μ V+SA constructs may be the result of higher EC loading. If I get the methods section right, there was bulk seeding of 3 million ECs in the SA constructs and additional perfusion seeding of the micro-channels with 10 million cells in the μ V+SA group.

We apologize for the possible confusion in the methods section for cell density of our constructs. We expect the cell density is very similar in the two constructs. We have added the following clarification, highlighted in the method section on Page 19-20: 1) The patterned microchannels were seeded with 10 μ L of hESC-ECs at density of 10 million/mL, therefore, 0.1 million cells in total seeded. 2) Seeded cells were allowed to attach the luminal surface then media was perfused through the microchannels to remove the non-attached cells in the reservoir or lumen. Based on the geometry of predefined microchannel network, we estimate 50,000 cells were maintained in the lumen during culture and implantation. 3) In μ V+SA condition, the volume of microchannel networks is perfused with media, which in SA only condition, however, would be occupied with cells in gels with 3 million/mL ECs. This volume accounts for roughly 40,000 cells in the SA only condition. With the total cell number in each construct of about 150,000 cells, the difference of 10,000 cells per construct, is not expected to have dominant effects responsible for high-fold changes in vessel density and perfusion.

EC survival after bulk seeding appeared to be a function of distance from the fabricated micro-channels. This is surprising given a construct thickness of 100 μ m for the in vitro studies. Life-dead stains or time-course assessment of apoptosis would help to better understand the differences in EC survival

We apologize for the lack of clarity of construct size. Our construct is 1 mm thick, whereas the vessel diameter is 100 μ m, embedded within the construct. We have now made it clear on Page 19 in main text and Page 4 in the supplementary information. In the revision, we have performed dead-stains for the μ V+SA construct, supplemented in fig. S2I, and showed that the reviewer is correct: ECs survival depends on the distance to the vessel perfusion.

Several groups have demonstrated vascularization of tissue grafts. The study would be markedly improved if enhanced survival of co-seeded cardiomyocytes could be demonstrated.

We agree with the reviewer that the ultimate goal of enhanced vascularization is to enhance survival of cardiomyocytes and host integration in both vasculature and electrical coupling. However, there are multiple steps needed to reach the goal. Though several groups have demonstrated vascularization of tissue grafts, no previous work has quantitatively measured perfusion and blood velocity in tissue grafts, not to mention in an infarcted heart. These serve as

outstanding challenges in the field of tissue engineering and regenerative medicine. Our work here addressed these challenges, and presented the first demonstration of improved perfusion in an infarcted heart model system using quantitative measurement. This presents the unambiguous evidence for the role of vascular patterning in enhancing vascular integration.

The true benefit of the micro-fabrication approach is not clear if for the most part vascularization is from endogenous cell sources.

To understand in more details, we have performed RNAseq analysis in the revision and found that $\mu\text{V}+\text{SA}$ constructs, compared to SA only condition, have upregulated hypoxia, glycolysis, $\text{TNF}\alpha$ signaling, MTORC1 signaling, and epithelial to mesenchymal transition. μV only constructs have similar trends as $\mu\text{V}+\text{SA}$ constructs when compared to SA only, though with less degree of change. When implanted in vivo, $\mu\text{V}+\text{SA}$ constructs had high cellular infiltration in the tissue grafts, and led to rapid vascularization, whereas μV only or SA only constructs had minimal vascularization. These data suggest that $\mu\text{V}+\text{SA}$ constructs have fundamentally advanced vascular development, supported by in vitro anastomosis results, RNAseq analysis and host integration results in vivo. We have included these RNAseq results on Page 10-11 and Fig. 3 and Supplementary Fig. S5, and discussion on Page 16.

The claim of having fabricated highly vascularized constructs should be toned down. Vascular density, mainly due to capillary ingrowth, remains rather low in vivo. As to the in vitro data, it is important to differentiate between cell-seeded micro-channels and bona fide vessels or capillaries. For the most part it is the channels that contribute to perfusion and not the EC sprouts.

We have toned down the language of highly vascularized constructs, as suggested by the reviewers. In the supplementary data, we have now showed vessel sprouts or connection with the bulk seeded ECs reduced overall flow resistance in vitro. Using RNAseq analysis, we have now found that perfusable microvessels together with self-assembled vessels ($\mu\text{V}+\text{SA}$ constructs) have significant increase of function in endothelial cell proliferation and vascular tissue development. In vivo, μV only constructs did not have rapid integration and perfusion at day 5, which we have now included in Supplementary Fig. S7A. Therefore, ECs in the surrounding matrix are important for further vascular tissue development, and contribute significantly to the host integration in vivo. We have included these results on Page 11-13.

Minor concerns:

The title should be revised to not overstate the key finding, which is the engineering of a blood perfusable microchannel system and its application to enhance graft vascularization by endogenous ECs. Evidence for outcome relevant anastomoses is not provided.

We have now revised the title to “Patterned human microvascular grafts enable rapid vascularization and increase perfusion in infarcted hearts”.

Figure legends need to be carefully revised. The provided information is in some legends insufficient to fully understand the panels, e.g., Fig 1a and SFig 2a, and sometimes lacking completely, e.g., Fig 2G-H. In Fig. 2B, arrows cannot be distinguished well.

We have now added more details in Figure legends.

Data on vessel density in the infarct area and a comparison to graft vessel density would be informative.

We have now included quantification data of vessel density in the infarct area and presented in Supplementary Fig. 8E.

Please, provide control lectin labelling data to confirm species specificity.

We have included the control lectin labelling data (positive control for GSL1, negative controls for UEA 1 and mTm-hESC-ECs) in Supplementary Fig. 8A.

Fig 4 panels C and D: a clarification is needed as to the differences in human vessel count.

We have added more information in figure 5 (figure 4 in the previous version) legends for clarification.

Reviewer #2

This is an extension of work conducted by this group for some time. The work presented is not conceptually novel as this group and others (Sasagawa T et al. J Tiss Engineer Regen Med 2016; Kang K-T et al. Blood 2011) have shown that in vitro patterned bioengineered vascular constructs can be shown to generate vessels in vivo when implanted into various tissues. This present work is an incremental advance to show that patches of prevascularized patterned collagen implants with bulk seeded hESC-EC demonstrate greater anastomosis with vessels in host injured tissues. The speed with which these implanted pre-patterned vascular structures connected with host vasculature is similar to prior reports of 3-5 days. Unfortunately, there are no mechanistic insights into how this type of implant is working to enhance vascularization of the ischemic tissue, and surprisingly despite improved blood flow, there is similar levels of ischemic heart damage and inflammatory cell infiltrate. This suggests that the accelerated perfusion of the tissue may not enhance the recovery of the damaged myocardium.

We thank the reviewer for bringing up the literature. In part, we agree, the vascularization concept is not conceptually novel since this is the ultimate goal for the field of vascular engineering and regenerative medicine. Toward this goal, however, we would like to emphasize that no existing work has quantitatively measured perfusion and blood flow in tissue grafts. Though previous studies have shown improved vascularization, it is not clear to what extent perfusion was affected. To rationalize the engineering design, we believe quantified measurement of blood perfusion in grafts and hosts will provide more informative outcomes towards optimization. Here, our study addressed these challenges by incorporating a new optical imaging tool, and demonstrated physiologic tissue perfusion in microfabricated constructs, and therein lies the novelty and impact of this study.

With regards to enhancing recovery (similar levels of ischemic heart damage), it is important to note that we are implanting the constructs on the epicardial surface to determine the effects of microfabrication on **graft** perfusion in a myocardial infarct model. This is not a neovascularization study aimed at directly reducing infarct size. Rather the similar infarct size and inflammatory response is an important control to ensure that the benefits we see are not due to skewed levels of injury as a result of varied surgical practices (we have shown this aspect in the supplementary Fig. 6).

To dig further on the possible mechanism differentiating μ V+SA constructs from SA only constructs, we have performed RNAseq analysis in the revision for the three construct

conditions: SA only, μ V only and μ V+SA constructs at the conditions prior to implantation. We found that μ V+SA constructs, compared to SA only condition, have upregulated hypoxia, glycolysis, TNF α signaling, MTORC1 signaling, and epithelial to mesenchymal transition. μ V only constructs have similar trends as μ V+SA constructs when compared to SA only, though with less degree of change. We have included these results on Page 10-11 and Fig. 3 and Supplementary Fig. S5. We believe the addition of this piece of data provides further innovation in revealing the role of perfusable vessels in supporting vascularization and increasing the host perfusion into the graft.

Reviewer #3:

Major Comments:

1. The authors need to more specifically state what they are testing in Figure 1. What is their hypothesis and what conclusions are they drawing from the data. The purpose of showing a comparison between SA alone and μ V+SA is not clear since there are no significant differences between the two conditions in E. Also, the colors used in G are misleading because the same colors are used for different conditions in E.

The goal in Figure 1 is to show that the bulk seeded ECs form anastomotic connections with patterned perfusable microvessels to establish greater vascular density and higher perfused area. The purpose of showing SA in the matrix surrounding μ V was to show the vessel anastomosis between micropatterned network with the self-assembled one in the matrix. The major difference between SA alone and μ V+SA is that SA alone was not perfusable whereas μ V+SA constructs were perfusable throughout the predefined network. We have edited the results section for Figure 1 for more clarification. We have also changed the color of lines in G to avoid confusion.

2. Along those same lines, there is little continuity between the in vitro and in vivo work. In vitro, the authors show a comparison between the micro-vessels (μ V) alone and the micro-vessels plus self-assembled lumens (μ V+SA), but in vivo the authors are comparing SA alone and μ V+SA. Please clearly state the rationale/hypothesis for these experiments? Also, it appears that the μ V graft alone might be sufficient to induce perfusion into the graft since SA alone has no flow in vivo? The authors need to include the μ V alone condition so that we can make a conclusion about the necessity of these two cell types for the observed effect.

We thank the reviewer for the suggestions. We have presented three conditions in the paper: SA only, μ V only, and μ V+SA constructs. We hypothesized that perfusion and patterned microvessel network enhance the vascular development and tissue function. Our goal was to combine the lithographically defined microvessel network with cell self-assembly into a highly vascularized construct. From in vitro imaging and perfusion analysis, we agree with the reviewer that there is not significant difference on the vascular perfusion, however, vascular density was significantly higher for μ V+SA vs μ V only. Furthermore, μ V only constructs did not induce rapid vascularization after implantation into the infarcted hearts for 5 days (we have now included the representative imaging data in Supplementary fig. S7A, and added text on Page 12).

To further understand the difference between the three constructs, we performed RNAseq analysis in the revision for the three construct conditions: SA only, μ V only and μ V+SA constructs at the conditions prior to implantation. We found that μ V+SA constructs, compared to SA only condition, have upregulated hypoxia, glycolysis, TNF α signaling, MTORC1 signaling,

and epithelial to mesenchymal transition. μ V only constructs have similar trends as μ V+SA constructs when compared to SA only, though with less degree of change. We have included these results on Page 10-11 and Fig. 3 and Supplementary Fig. S5. Our data suggested the necessity of two types of vessels (patterned perfusable vessels & self-assembled tubes in the matrix) and their connection or anastomosis promoted the host integration and rapid vascularization.

3. What is the evidence that the cells are endocardial-like. There needs to be endocardial marker expression shown for the cells, not just those of endothelial cells, if this claim is to be made.

We have defined these cells as endocardial-like cells based on their high NFATC1 expression, from our previous work (Palpant et al, Development, 2015). We have now included NFATC1 staining on these cells in Suppl Fig. S1D. We agree with the reviewer that endocardial-like ECs may be overclaimed using the single marker. We have now simplified the definition as hESC-ECs (hESCs derived endothelial cells) in the revised manuscript.

4. Figure 1B and C are said to show angiogenic activity. There are no parameters of angiogenesis shown in these panels. These same Figures are also said to show lumens, which are not shown in the micrographs.

We have now changed the language of angiogenic activity to “sprouted extensively into the bulk matrix” on Page 6, and changed the figure citation to the supplementary figures where the zoomed views are shown.

5. For video S1, it is not obvious that these are "highly integrated microvascular networks". Yes, the green cells are intermixing with the red, but there needs to be more evidence of a network to make this conclusion. What do the authors mean by network? It usually describes a network of perfused interconnect blood vessels. Perhaps show 3D views of the confocal image and have to movie rotate it instead of scrolling through?

We thank the reviewer for the comments and apologize for the lack of clarity. We have now changed the language and focused on the obvious evidence: The newly formed self-assembled tubes integrated with angiogenic sprouts from the μ V network and formed numerous anastomotic connections. We did not find a better view when rotating the video, therefore, left video S1 unchanged. To add more evidence, we have now added a z-projection image set in Fig. S2E to show the connections and highly integrated microvessel networks.

6. The authors conclude that: “these data suggest that the localized vascular remodeling and anastomotic events decrease the vascular resistance allowing for more efficient perfusion through endothelial sprouts”. More evidence needs to be provided to make this conclusion, including a measurement or modeling of resistance in the networks.

We have included COMSOL models of flow through the vessel network comparing a sprouted network and a vessel network with baseline structure. We showed that sprouts from the vessel network will lead to higher flow at the same pressure drop between inlet and outlet, therefore, lower flow resistance. This leads to 25% increase in the maximal flow velocity in the sprouted vessel network compared to the original vessel geometry in our COMSOL models. We have added these results on Page 8 and Supplementary fig. S3D.

7. In Figure 1D, evidence for a tube-like vessel anastomosing with the patterned tube needs to be stronger.

We have added more example z-projection images in Fig. S2E to show the tube-like vessel anastomosis with the micropatterned vessels.

8. There is no evidence of rapid anastomosis in vivo. This makes the title of the manuscript misleading because it implies that there is rapid anastomosis in vivo as well. Experiments need to be performed to provide support that rapid anastomosis plays a role in the increased perfusion they observe in vivo.

We have changed the title to enhance clarity. We agree with the reviewers that we do not have clear evidence to show rapid anastomosis in vivo. The μ V+SA constructs showed greater lumen density and increased perfusion in comparison to SA only when implanted in vivo. We also showed μ V+SA groups contain chimeric vessels that are comprised of host ECs and human ECs suggesting anastomosis in vivo whereas SA groups have human EC lumens and host vessels that are more distinct from each other. To add more evidence, in the revision, we have performed RNAseq (Page 10-11, Fig. 3) and showed μ V+SA constructs have unique gene expression clusters, particularly regarding hypoxia, glycolysis, and vascular development, compared to μ V only or SA only constructs, suggesting the anastomosis in vitro are important for the increased perfusion in vivo. In addition, we have added data for μ V only constructs (Supplementary fig. S7A), and showed the lack of perfusion and rapid vascularization in vivo.

9. The thrombogenic experiments need positive (activated ECs?) and negative (non-adherent surface only?) controls. There is some adhesion. Are these physiologically relevant levels?

We have added more experimental results for blood perfusion in Fig. 2G-H and Supplementary fig. S4C-F. We have shown that HUVEC-formed microvessels, as negative control, showed minimal platelet adhesion (Supplementary fig. S4C). hESC-EC microvessels have higher platelet adhesion than HUVEC microvessels, and both are expected to be higher than the physiological relevant level since these vessels have never been perfused with blood before. We further showed hESC-ECs formed microvessels have significantly higher platelet adhesion after vessel activation by PMA or IL-1 β . These data suggest that hESC-ECs microvessels can be activated by the cytokines and drugs and become thrombogenic.

10. The authors need to explore what aspect of the vascular graft contributes to the increased perfusion in vivo. Most of the perfused vessels are from the host – is that solely responsible for the total increase in perfusion observed. Why are the SA cells included then? Again, looking at the μ V graft alone would help.

We have added μ V only in vivo data in the revision in Supplementary figure S7A. We found that μ V only microvessels led to minimal graft perfusion, similar as SA only constructs, and much less than μ V +SA constructs. Clearly, SA cells are important contributors to the increased vascularization and host integration. To understand further, we have performed RNAseq analysis for the three construct conditions: SA only, μ V only and μ V+SA constructs after three days of culture in vitro (the same condition as prior to implantation). We found that μ V+SA constructs, compared to SA only condition, have upregulated hypoxia, glycolysis, TNF α signaling, MTORC1 signaling, and epithelial to mesenchymal transition. μ V only constructs have similar trends as μ V+SA constructs when compared to SA only, though with less degree of change. We have included these results on Page 10-11 and Fig. 3 and Supplementary Fig. S5.

11. Additional experiments showing improved outcome (i.e. increased ejection fraction or cardiac function) would benefit the manuscript.

We agree that evaluating cardiac function would be the next step of the work. The ultimate goal of enhanced vascularization is to enhance survival of cardiomyocytes and host integration in both vasculature and electrical coupling. However, there are multiple steps needed to reach the goal. Outstanding challenges exist in the field of tissue engineering and regenerative medicine to quantitatively measure perfusion and blood velocity in tissue grafts, not to mention in an infarcted heart. Our work here addressed these challenges, and presented the first demonstration of improved perfusion in an infarcted heart model system using quantitative measurement. We believe this is an important step forward for the field to provide rationalized outcome towards optimization of engineering designs.

Minor Comments:

1. I could not find how the cell density was normalized between the μ V-SA and SA constructs – is the effects seen only an effect of differences in cell density? Clarification is needed to ensure that the total number of cells is the same in μ V-SA, μ V, and SA constructs.

We have added the information for cell density in the matrix and lumen on Page 19-21 and added cell number estimate in the method section. The cell number for μ V+SA and SA constructs are very similar.

The patterned microchannels were seeded with 10 μ L of hESC-ECs at density of 10 million/mL, therefore, 0.1 million cells in total. Seeded cells were allowed to attach the luminal surface then media was perfused through the microchannels to remove the non-attached cells in the reservoir or lumen. Based on the geometry of predefined microchannel network, we estimate 50,000 cells were maintained in the lumen (μ V only constructs) during culture and implantation. In μ V+SA condition, the volume of microchannel networks is perfused with media, which in SA only condition, however, would be occupied with cells in gels with 3 million/mL ECs. This volume accounts for roughly 40,000 cells in the SA only condition. With the total cell number in each construct of about 150,000 cells, the difference of 10,000 cells per construct when comparing μ V+SA and SA only conditions, is not expected to have dominant effects responsible for high-fold changes in vessel density and perfusion.

2. The authors should provide evidence of why they are concluding that cell death is responsible for decreased EC density 600 μ m away from the patterned vessel wall in vitro.

We have performed dead stains and showed significant cell death at 600 μ m away from the patterned vessel wall in vitro.

3. Could the authors provide the figure/data for the statement: “10-fold difference between the total number of perfused vessels and the number of perfused human vessels”?

Figure 5D showed this information, we had detailed number information on Page 13 and summarized in the discussion section for this statement.

4. Typo – “(0.72 mm/s in in 20-30 ...”

We have corrected it.

5. No error bars in figure 2D and supplementary figure 4A

We have added more replicates for flow speed measurements and included them in figure 2D and supplementary figure S3A (figure S4A in previous version).

6. One of the images should be labeled “XY” in figure 2H, not both “XZ”

We have changed the figure 2G to show projections of microvessels at XY planes only for both control and activated conditions. .

7. The caption for figure 1A and 1D are confusing because they have subsections (i-iv), but the caption doesn't explain what the differences are. It is unclear from the caption that they are related (or not related). Also consider using different numbering for 1C since it is not related to 1A and 1D (i.e. a,b,c,d). A similar issue with crossover in numbering comes up in figure 2B (since it's not related to 2A).

We have changed the figure caption and subsection labeling to enhance the readability.

8. The confocal image for figure 1F seems to be tilted. Is there specific reason for this? Otherwise, just show the XY plane. Also, it is confusing where the interstitial collagen is – is it relevant to the figure?

Figure 1F is a 3D reconstructed image, as stated in the figure legend and in the results section. The interstitial collagen label is simply to orient the reader as to the geometry of the region within the image. We have added more information in the legend and correlated it to video S1.

9. Supplementary figure 3A-C should have the markers labeled on the figure itself even if the names are in the caption. In addition, adding the distance labels to the figure would make it clearer.

We have added the distance labels in the figures (Supplementary figure S2E-G in the revision) to enhance readability.

Reviewers' Comments:

Reviewer #1:

Remarks to the Author:

This is a revised manuscript by Redd and colleagues on enhanced myocardial vascularization after implantation of precapillarized collagen patches. Quantitative imaging of tissue perfusion using innovative OCT methods is a clear strength of the study and I concur with the authors that this type of rigorous assessment is indeed lacking. The finding of enhanced capillary flow after implantation of $\mu\text{V}+\text{SA}$ patches is encouraging because it may help to enhance cardiomyocyte survival if co-implanted. The authors should be in the position to test this hypothesis and strengthen their study.

Major concern:

The authors should provide data on the survival of cardiomyocytes embedded in $\mu\text{V}+\text{SA}$ vs. μV or SA at the chosen study endpoint, i.e., 5 days after implantation. Without this data it remains quite unclear whether the proposed strategy would indeed be beneficial, i.e., supportive as to cardiomyocyte survival.

Minor issues:

Video 5 did not present well on my screen.

SFig 3: the column for the day 7 $\mu\text{V}+\text{SA}$ data should not contain an error bar if from only 2 samples

Reviewer #3:

Remarks to the Author:

The authors sufficiently addressed concerns regarding the title of the manuscript and provided additional RNAseq data revealing expression differences when SA is added to μV . Specifically that cells activate pro-inflammatory gene expression only when μV and SA cells are cultured together. Finally, they provide evidence through some CFD modeling that flow is increased within the construct.

Remaining comments

1. An important control was to show that the μV only group is not perfused, making inclusion of SA an important component. Because this point is central to their claims, the authors might consider adding the data shown in Figure S7 (no perfusion with μV only) into main Figure 4.
2. The authors explain that there are 150,000 and 140,000 cells in the in the $\mu\text{V}+\text{SA}$ vs. SA only conditions. However, a better explanation is required on how they arrive at these numbers. From the explanation in the reviewers response, I would calculate that $\mu\text{V}+\text{SA}$ have 90,000 cells and SA only have 40,000. Please better explain how many cells are included into each construct.
3. Please elaborate on the COMSOL parameters used to model the flow within the construct (i.e. wall properties, Reynolds number, etc.) in the methods to enhance reproducibility.
4. Figure S7A should have labels for the two rows of image (μV implant 1 and 2 for example) on the figure itself.

Response to Reviewers

Reviewer #1

We thank the reviewer for the recognition of the innovative approaches and interesting findings in our work, and helpful suggestions for improving the manuscript. In response we have now included additional data, which we hope alleviates the reviewer's concerns and improves the clarity and quality of our manuscript.

Major concerns:

The authors should provide data on the survival of cardiomyocytes embedded in $\mu\text{V}+\text{SA}$ vs. μV or SA at the chosen study endpoint, i.e., 5 days after implantation. Without this data it remains quite unclear whether the proposed strategy would indeed be beneficial, i.e., supportive as to cardiomyocyte survival.

We thank the reviewer for this suggestion and have addressed these concerns by performing an additional *in vivo* experiment in which we implanted $\mu\text{V}+\text{SA}$ and SA constructs that contained additional hESC-derived cardiomyocytes. We found that $\mu\text{V}+\text{SA}$ cardiac constructs had a significantly higher density of cardiomyocytes that remained in the graft with consistently higher vascular density at day 5 post-implantation compared to SA cardiac constructs. Furthermore, nearly all cardiomyocytes in the graft stained negative in a TUNEL assay. Although the average graft size was larger in $\mu\text{V}+\text{SA}$ cardiac constructs, the trend was not statistically significant. Because the SA cardiac constructs had more bulk volume and cardiomyocytes during initial fabrication, the slightly smaller graft size after implantation suggests our $\mu\text{V}+\text{SA}$ constructs better supported cardiomyocyte remodeling and engraftment than SA constructs. We have now reported these results on Page 15-16 and Fig. 6 and Supplementary Figure 9.

Minor concerns:

Video 5 did not present well on my screen.

We have now included an additional video at less zoomed view to show the mild platelet binding along the main vessel networks. We also kept the original video, though slightly lower resolution, to show the platelets moving through the small sprouts without significant binding in normal culture conditions.

SFig 3: the column for the day 7 $\mu\text{V}+\text{SA}$ data should not contain an error bar if from only 2 samples

We have removed the error bar from the day 7 $\mu\text{V}+\text{SA}$ data in Supplementary Figure 3.

Reviewer #3:

Remaining Comments:

1. An important control was to show that the μV only group is not perfused, making inclusion of SA an important component. Because this point is central to their claims, the authors might consider adding the data shown in Figure S7 (no perfusion with μV only) into main Figure 4.

We agree that the μV only data is an important control, however, we have kept this data in the supplementary figure (Supplementary Figure 7), due to the different OMAG schemes utilized for image scanning. Due to the lack of detection of perfusable vessels in the μV only grafts, we have utilized the original less-quantitative OMAG imaging scheme for flow measurement in this condition, and did not repeat the measurement for quantitative OMAG velocity (OMAG-V) data at the same setting with all other conditions shown in Fig. 4. We hope the data presentation and reference to this supplementary figure provide sufficient comparison and attention to this control condition.

2. The authors explain that there are 150,000 and 140,000 cells in the in the μV +SA vs. SA only conditions. However, a better explanation is required on how they arrive at these numbers. From the explanation in the reviewers response, I would calculate that μV +SA have 90,000 cells and SA only have 40,000. Please better explain how many cells are included into each construct.

We apologize for the confusion. We believe our number estimation reflects the accurate cell number in the constructs. In all our constructs, the cell density in the bulk matrix has been remained at 3 million cells/mL (3,000 cells/mm³) at fabrication. An original SA construct has dimensions of 8 mm in diameter and 1 mm in thickness, leading to a volume of 50 mm³. An The μV +SA constructs have volume of approximately 40 mm³ due to the luminal space for endothelial cells seeding and culture under perfusion. Additional cells were seeded into the lumen and residual cells were washed away after cells were attached, leading to roughly 50,000 cells on the lumen.

We have also subtracted the volume (~3 mm³) taken out by the 2 mm biopsy punch (two half punches on each side) for both constructs to match the practical experimental conditions upon implantation. We have added more volume calculations in the methods section for further clarification.

3. Please elaborate on the COMSOL parameters used to model the flow within the construct (i.e. wall properties, Reynolds number, etc.) in the methods to enhance reproducibility.

We apologize for missing the COMSOL parameters. We have used COMSOL software and solved two-dimensional Navier-Stokes equation for the fluid flow assuming steady state and laminar flow. The fluid is assumed to be Newtonian with the same property as water. The network geometry was selected the same as Fig. 1-2, with diameter of 100 μm . The flow rate was computed based on the pressure drop applied between the inlet and outlet, which is set to be 100 Pa (inlet: 100 Pa, and outlet: 0 Pa relative to the atmosphere). The sprouted network geometry was expanded upon the original one with roughly 25% increase of the original vessel area with two long sprouts at the diagonal corner of the network, as we observed in the experiments. The same pressure drop was applied between the major flow inlet and outlet, in addition, the ends of two long sprouts were also set to have the same outlet pressure. The extra fine mesh was used for numerical simulation, and convergence was reached with tolerance < 10⁻⁶ for the change of velocity and pressure in the whole field. We have now added the related parameters in supplementary figure 3 legend on Page 3-4 of supplementary material.

4. Figure S7A should have labels for the two rows of image (μV implant 1 and 2 for example) on the figure itself.

Labels including implant type and number have been included in Supplementary Figure 7A.

Reviewers' Comments:

Reviewer #1:

Remarks to the Author:

The authors have performed the suggested experiment and are presenting new data in support of the claim that cardiomyocyte survival is enhanced in the μ V supported constructs. I have no further concerns.